# Mamba Only Glances Once (MOGO): A Lightweight Framework for Efficient Video Action Detection

**Yunqing Liu[1], Nan Zhang[1], Fangjun Wang[1], Kengo Murata[2], Takuma Yamamoto[2],**
**Osafumi Nakayama[2], Genta Suzuki[2], Zhiming Tan[1]\***
[1]Fujitsu R&D Center    [2]Fujitsu Research Japan

## Abstract

Mamba, a lightweight sequence modeling framework offering near-linear complexity, presents a promising alternative to Transformers. In this work, we introduce MOGO (Mamba Only Glances Once), an end-to-end framework for efficient video action detection built entirely on the Mamba architecture. In MOGO, our newly designed Mamba-based decoder can even use just one Mamba layer to effectively perform action detection. It uses neither Transformer structures nor RCNN-like methods for proposal detection. Our framework introduces two key innovations. First, we propose a pure Mamba-based encoder-decoder architecture. The encoder processes cross-frame video information, while the decoder incorporates two novel Mamba-based structures that leverage Mamba's intrinsic capabilities to detect actions. Theoretical analysis and ablation experiments confirm their synergy and the necessity of each structure. Second, we design a video token construction mechanism to improve the model's performance. The token importance block can ensure that the retained token information is highly relevant to the predicted targets. These two innovations make MOGO both efficient and accurate, as demonstrated on the JHMDB and UCF101-24 benchmark datasets. Compared to SOTA action detection methods, MOGO achieves superior performance in terms of GFLOPs, model parameters, and inference speed (latency) with comparable detection precision. Additionally, it requires significantly less GPU memory than some SOTA token reconstruction methods. Code is available at https://github.com/YunqingLiu-ML/MOGO.

## 1 Introduction

The goal of video action detection is to localize and classify actions within video sequences, requiring models to capture spatiotemporal dependencies. Recently, Sia and Rawat [1] address this by introducing a lightweight encoder-only model for open-vocabulary detection. Other advancements in this field, such as DETR [2] and TubeR [3], have relied on Transformer-based architectures. These models leverage the attention mechanism to model long-range dependencies across video frames, achieving SOTA performance in capturing temporal context and action semantics. However, the self-attention mechanism incurs quadratic computational complexity with respect to sequence length, making it computationally expensive for long video sequences. For instance, the methods proposed in [4] and [5] achieved significant performance improvements but are reported to have relatively high GFLOPs. This inefficiency makes such models less practical for resource-constrained environments.

The Mamba framework [6] emerges as a promising alternative to Transformers. Unlike the attention-based paradigm, Mamba employs state-space modeling to achieve near-linear complexity, offering a scalable solution for processing spatiotemporal data. Recent adoptions, such as VideoMamba [7] for video classification and MS-Temba [8] for temporal action localization, demonstrate Mamba's

---

*Corresponding author: zhmtan@fujitsu.com.

39th Conference on Neural Information Processing Systems (NeurIPS 2025).

efficiency in handling sequential data. However, its potential remains underexplored in the context of video action detection.

To develop an efficient and effective framework for video action detection, we proposed a purely Mamba-based framework. The motivation of our design is detailed in Figure 1. The Transformer depends on self-attention and cross attention to capture the dependency relationships. In contrast, Mamba utilizes linear transformations to project queries into target spaces and employs selective copying to efficiently model cross-sequence information. Building on this, we propose MOGO (Mamba Only Glances Once), an end-to-end framework for efficient video action detection. Unlike SOTA methods that incorporate external structures such as RCNN for region proposals [4] or Transformer components like ViT [9], MOGO eliminates the need for these additional modules. Our framework introduces two key innovations:

(1) Pure Mamba-based Architecture. Our designed decoder processes learnable queries and video information tokens through a streamlined pipeline of EQ-Mamba, QVI-Mamba, and an FFN as shown in Figure 2. Ablation studies confirm the synergistic necessity of each module.

(2) Video Token Construction Mechanism. As shown in Figure 3, this module computes importance scores for encoder tokens across spatiotemporal frames. Coupled with an target-guided loss function, this mechanism reduces redundancy and enhances token relevance, further boosting efficiency.

Figure 1: **Motivation of MOGO.** Comparison between Transformer and Mamba complexity inspires our framework design. By *Mamba-izing* the Transformer's (1) attention architecture, our framework enables (2) linear transformation for single-sequence modeling and (3) selective copying mechanism for cross-sequence modeling.

The whole method's main strengths are computational efficiency and a token-importance block, as demonstrated on the JHMDB [10] and UCF101-24 [11] benchmark datasets. Compared to SOTA action detection methods, MOGO achieves superior performance in terms of GFLOPs, model parameters, and inference speed (latency) with comparable detection precision. Additionally, it requires significantly less GPU memory than some SOTA token reconstruction methods.

## 2 Method

### 2.1 Overall MOGO Architecture

Our proposed MOGO framework introduces an end-to-end action detection pipeline built entirely on a Mamba-based architecture, as illustrated in Figure 2. Specifically, it detects actions on a designated keyframe within a short video clip while using the remaining frames as temporal context. Because predictions are made on a single keyframe, some non-keyframe tokens may be redundant or weakly informative, which motivates selective token retention and importance modeling, as illustrated in Figure 3. Unlike SOTA methods that incorporate external structures such as RCNN for region proposals [4] or Transformer components like ViT [9], MOGO leverages only Mamba's inherent capabilities, with a primary goal of minimizing computational overhead while maintaining robust detection performance. The MOGO architecture comprises several key components, and the main elements include: Mamba encoder, Mamba decoder, token importance module, and loss function.

### 2.2 Pure Mamba-based Architecture

Our proposed method adopts an end-to-end purely Mamba-based architecture as shown in Table 1:

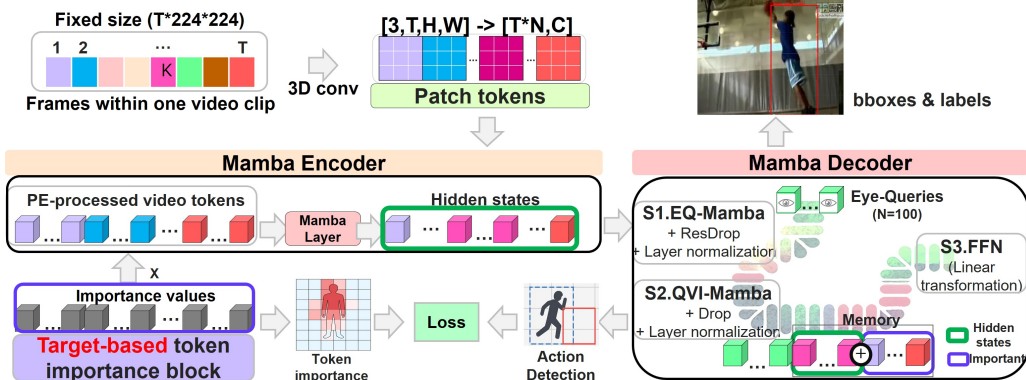

Figure 2: **Overview of the proposed structure.** The encoder processes PE-embedded video tokens through Mamba layers, accompanied by a target-based token importance block that highlights informative regions for action detection. The decoder then operates on two Mamba-based modules followed by a linear layer. Illustration corresponds to batch size = 1. (EQ-Mamba: Eye Query-based Mamba, QVI-Mamba: Query-Video Information-fused Mamba, PE: Positional Embedding, ResDrop: Residual connection with Dropout, Drop: Dropout, K: Key frame.)

**Encoder.** Leveraging the pretrained models [7], our encoder is designed to extract comprehensive feature maps from input video frames. Following the patch embedding, which transforms the input clip into a sequence of patches, we incorporate spatial positional embeddings and temporal positional embeddings to model inter-frame dependencies across the video sequence. Then we incorporate a bidirectional Mamba setup. The encoder comprises $l_e$ layers, which is evaluated through ablation studies in Section 3.4.

Table 1: **Tensor shape flow across the MOGO architecture.** $M$ contains keyframe tokens and important non-keyframe tokens.

| Stage | Tensor | Shape / Description |
|---|---|---|
| Input | $x$ | $[B, 3, T, H, W]$ |
| Patch Embedding | $\text{Conv3D}(x)$ | $[B, C, T, H', W']$, where $C = \text{embed\_dim}$ |
| Flatten and Rearrange | $x$ | $[B, T \cdot N, C]$, $N = H' \times W'$ |
| Add Positional Embedding | +PE, +TPE | $[B, T \cdot N, C]$ |
| Token Importance Block | $\text{MLP}(x)$ | $[B, T \cdot N, 1]$ |
| Apply Mask | $x \cdot \sigma(\text{MLP}(x))$ | $[B, T \cdot N, C]$ |
| Mamba Encoder (depth = $l_e$) | $\text{Block}_1 \rightarrow \cdots \rightarrow \text{Block}_{l_e}$ | $[B, T \cdot N, C]$ |
| Mamba Decoder (depth = $l_d$) | $\text{Block}_1 \rightarrow \cdots \rightarrow \text{Block}_{l_d}$ | – |
| - Decoder Queries | $Q = \text{Embedding}(N_q)$ | $[N_q, B, C]$ |
| - Decoder EQ-Mamba | $Q'$ | $[N_q, B, C]$ |
| - Decoder QVI-Mamba | $\text{Concat}(Q', M)$ | $[N_q + M, B, C] \rightarrow [N_q, B, C]$ |
| - Decoder FFN | $F$ | $[N_q, B, C]$ |
| Prediction Heads | Class: $\text{Linear}(F)$ | $[B, N_q, \text{num\_classes} + 1]$ |
| | Box: $\text{MLP}(F)$ | $[B, N_q, 4]$ (normalized) |

**Decoder.** Our decoder, as depicted in Figure 2, uses the Mamba operator to process queries and integrate video information. Given an input sequence $x \in \mathbb{R}^{L \times d}$, the Mamba block computes its output $y \in \mathbb{R}^{L \times d}$ via a state-space model:

$$
\begin{aligned}
h_t &= \mathbf{A} \cdot h_{t-1} + \mathbf{B} \cdot x_t \\
y_t &= \mathbf{C} \cdot h_t
\end{aligned}
\tag{1}
$$

where $\mathbf{A}, \mathbf{B}, \mathbf{C} \in \mathbb{R}^{d \times d}$ are learnable transition and projection matrices, and $h_t$ is the hidden state at time step $t$. Compared with attention-based modules, this formulation enables linear-time complexity in sequence length $L$.

In the decoder design, Mamba serves two key functions: (1) linear transformation, and (2) feature extraction and sequence modeling based on selective copying [6]. EQ-Mamba primarily uses the first function, while QVI-Mamba exploits the second. Additionally, the concatenation of query and video information in the decoder draws inspiration from [7]. The decoder, consisting of $l_d$ layers, is evaluated through ablation studies in Section 3.4 and A.4. Specifically, EQ-Mamba operates solely

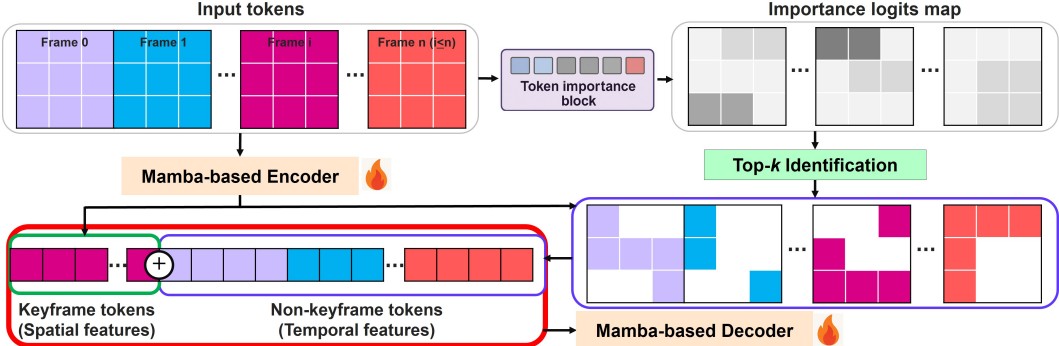

Figure 3: **Structure of the video token construction mechanism.** This module enables the decoder to obtain sufficient and effective information for action detection. Sufficient information is ensured by incorporating the complete spatial features of the keyframe (prediction frame), while redundancy reduction is achieved by filtering encoder tokens through token importance block. Importance scores are computed for each token in the encoder, retaining only the top-$k$ highest-scoring tokens to supply temporal information. These tokens are then combined with a fixed keyframe memory portion, which preserves semantic context.

on the query embeddings, projecting and refining them within their own latent space. This process resembles self-attention in its focus on intra-query relationships but relies on Mamba's state-space modeling instead of attention mechanisms, enabling efficient linear transformations of the query tokens. QVI-Mamba integrates the refined query embeddings with the encoder's video features (hidden states), concatenating them to create a mixed representation that captures both query-specific information and video context. Finally, the FFN processes the output of the QVI-Mamba block, further transforming the fused representations to predict bounding boxes and class probabilities.

The key difference between Mamba and Transformer architectures underpins our design choices. While Transformers serve as attention and correlation mechanisms between tokens, Mamba relies on its unique mechanism of information storage and linear transformation via state-space dynamics.

## 2.3 Video Token Construction Mechanism

Prior work [4] shows that employing a token-importance mechanism improves performance over processing all tokens. In this work, to optimize token utilization, we introduce a token importance module. This module selects the most relevant tokens from the encoder, which aggregates tokens from different spatiotemporal frames, to enhance the decoder's ability to capture salient features. In encoder, it employs an MLP to compute importance logits for each token, followed by a sigmoid activation to derive importance scores. It should be noted that this part consists of learnable parameters, which are dynamically adjusted based on the token importance loss. For details on the computation of the loss, refer to Section 2.4. The decoder's input tokens are derived by selecting high-scoring tokens based on their importance, resulting in a more informative and compact representation for processing, as shown in Figure 3.

Specifically, we compute importance scores for each token in the encoder, and then guide the token selection process in the decoder. To preserve the features learned from the large-scale pretraining model, we do not reduce the number of tokens in the encoder; instead, token importance serves only as an indicator at this stage. The importance scores are directed by a learnable mechanism. After calculating the importance scores, we perform token selection by picking the top $k$ percentage of tokens with the highest importance scores for each frame, keeping only the most informative tokens for further processing in the decoder. To further enhance this process, we combine a fixed portion of the memory, which contains all the keyframe tokens, with the selected tokens. The keyframes are crucial as they contain the accurate semantic information for the current frame. Next, we describe this process in tensor notation. For a batch of size $B$, the encoder processes a total of $T \times N_f$ tokens, where $T$ is the frame number, and $N_f$ is the number of tokens per frame. The token importance block assigns an importance score to each token (derived via the learnable MLP with sigmoid activation). Based on a top-$k$ selection strategy ($k$ is evaluated through ablation studies in Section 3.4.), we retain $\lfloor k \cdot N_f \rfloor$ tokens per frame. These selected tokens, totaling $N_s = \lfloor k \cdot (T \cdot N_f) \rfloor$, are concatenated with the keyframe tokens (fixed at $N_f$ per keyframe).

### 2.4 Loss Functions

Since Mamba does not possess an intrinsic attention-based importance mechanism as Transformers do, this section seeks to answer the question: what constitutes important or relevant tokens? We do so under the premise that the L2 magnitude of a token does not directly represent its actual importance. Our training objective comprises two parts:

**Detection Loss ($\mathcal{L}_1$):** We adopt a DETR-style loss to supervise the action detection, which integrates three components: classification, bounding box position, and overlap, assigned weights of $w_{\text{cls}}$, $w_{\text{box}}$, and $w_{\text{ovl}}$, respectively. A Hungarian Matcher is employed to align predictions with ground-truth targets. These weight values are ablated in our ablation experiments to optimize performance, as we do not directly adopt coefficients from prior DETR-based works (e.g., [2], [12]), given that our loss function is newly designed to suit the Mamba-based architecture. There's also a weight for the *no-object* class.

**Token Importance Loss ($\mathcal{L}_{\text{importance}}$):** To learn token importance scores, we construct a binary ground-truth mask $y_i^{(b)}$ for each token $i$ in sample $b$, indicating whether the token's center coordinates fall inside any ground-truth bounding box. Let $\alpha_i^{(b)}$ be the raw logit from the MLP for token $i$ in sample $b$. We compute the binary cross-entropy loss:

$$\mathcal{L}_{\text{importance}} = -\frac{1}{BN} \sum_{b=1}^{B} \sum_{i=1}^{N} \Big[ y_i^{(b)} \log(\sigma(\alpha_i^{(b)})) + \big(1 - y_i^{(b)}\big) \log\big(1 - \sigma(\alpha_i^{(b)})\big) \Big] \qquad (2)$$

where $B$ is the batch size, $N$ is the total number of tokens per sample, and $\sigma(\cdot)$ is the logistic sigmoid.

**Total Loss:** The final loss combines the two objectives:

$$\mathcal{L} = \mathcal{L}_1 + \lambda \cdot \mathcal{L}_{\text{importance}}, \qquad (3)$$

where $\lambda$ balances action detection performance and token importance learning, which is evaluated through ablation studies in Section 3.4.

Mamba lacks an attention map, unlike Transformers, which use attention scores to represent token importance. Transformers motivated us to design token importance. The proposed token importance block is not an off-the-shelf, plug-and-play module, but a learning-based, multi-stage system requiring end-to-end training. The key steps are:

(1) Encoder-side importance calculation. The encoder includes a trainable MLP that outputs a continuous importance value for each token, supervised by our custom loss.

(2) Token selection in the decoder. The learned importance scores are used to select important tokens from the non-keyframes, which are concatenated with the keyframe tokens as the decoder input.

(3) Loss design. The overall loss combines standard action detection loss and token importance loss. The optimal loss ratio is determined through ablation studies.

This system's effectiveness relies on the synergy of all its components and joint training.

## 3 Experiments

### 3.1 Experimental Setup

We evaluate our MOGO on three common datasets for video action detection: JHMDB [10], UCF101-24 [11] and AVA [13]. JHMDB contains 928 trimmed videos from 21 action classes. We follow the data annotation and procedure outlined in [14]. UCF101-24 comprises 3,207 videos spanning 24 sports classes. Following standard practice, we report performance on split-1. AVA is a large-scale benchmark and contains 299 15-minute videos, divided into 211k training clips and 57k validation clips. The results for AVA are shown in Section 4. We evaluate performance with mAP under an IoU threshold of 0.5 on NVIDIA A40 GPUs. Other implementation details are shown in Section A.3.

Table 2: **Efficiency analysis of MOGO.**

| Module | #Param | GFLOPs |
|---|---|---|
| Encoder | 73.996M | 101 |
| Query_embed | 57.6K | - |
| Decoder | 7.228M | 2.297 |
| Class_embed | 13.271K | 1.325e-3 |
| Bbox_embed | 0.667M | 6.659e-2 |
| Overall | 81.962M | 104 |

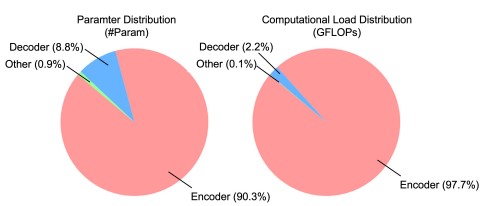

Figure 4: **Breakdown of parameter and computational distribution across components.** Based on the pretrained encoder, only a small amount of extra computation is added while still achieving good performance.

## 3.2 Efficiency Analysis

To assess the computational efficiency of MOGO, we measure the number of parameters, GFLOPs and inference speed and follow the code procedure in [7].

First, Table 2 details the parameters and GFLOPs for the model and its key modules. The entire MOGO model comprises 81.962 million parameters and incurs around 104 GFLOPs, reflecting a lightweight design compared to SOTA methods (see Section 3.3). The encoder dominates the computational load, accounting for roughly 97.7% of the total FLOPs as shown in Figure 4, as the encoder processes the 8-frame input using the pretrained Mamba-based backbone. In contrast, the decoder is significantly lighter, with 7.228M parameters and 2.297 GFLOPs, while auxiliary components like query_embed (57.6K parameters), class_embed (13.271K parameters), and bbox_embed (0.667M parameters) contribute minimally to the overall computational cost. This means that, based on the pretrained encoder, we only added a small amount of extra computation but still achieving good precision performance.

Second, we benchmark the inference speed using the throughput protocol defined in [4] with a video clip length of 8 frames. We run the model with 3 warm-up iterations and compute the average inference time over 10 additional runs. Our results show that MOGO achieves an average latency of 3.9 ms/img, corresponding to a throughput of 256 img/s. As shown in Table 3, this performance surpasses existing transformer-based methods: EVAD [4] (240 img/s on ViT-B), WOO [5] (176 img/s on ViT-B, 147 on SF-R101), and TubeR [3] (64 img/s). While EVAD reports a peak throughput of 334 img/s under certain alternative configurations, our approach remains highly competitive, even though the current version of Mamba's CUDA/C++ kernels still has limitations in parallel processing relative to Transformer implementations.

## 3.3 Comparison with SOTA Methods

Table 3: **Comparison of MOGO with SOTA methods.** GFLOPs are reported where available. Art.: Architecture, Thrp.: Throughput. C: CNN, T: Transformer, M: Mamba. dec.: decoder. Cv.: The original work only reported GFLOPs based on image-based RGB input, which has been converted here. J: JHMDB, U: UCF101-24.

(a) Comparison on efficiency.

| Model | Art. | Backbone | #Param | GFLOPs | Thrp. |
|---|---|---|---|---|---|
| SlowFast [15] | C | SF-R101-NL | - | 234×30 | - |
| MOC [14] | C | DLA34 | - | 235.2 [Cv.] | - |
| MaskFeat [16] | T | MViTv2-L | 218M | 2828 | - |
| EVAD [4] | T | ViT-L | 185M (dec.) | 737 | 153 |
| MeMViT [17] | T | MViTv2 | 52.6M | 620 | - |
| VideoMAE [18] | T | ViT-L | 305M | 597 | - |
| WOO [5] | T | ViT-B | 314M (head) | 378 | 176 |
| WOO [5] | T | SF-R101 | 314M (head) | 252 | 147 |
| EVAD [4] | T | ViT-B | 185M (dec.) | 243 | 240 |
| TubeR [3] | T | CSN-152 | - | 240 | 64* |
| **Ours (MOGO)** | M | Mamba-M | **82M** | **102–104** | **256** |

(b) Comparison on performance.

| Model | Art. | Backbone | Pre-train | f-mAP (J) | f-mAP (U) |
|---|---|---|---|---|---|
| EESSL [19] | C | I3D-CNN | - | 64.4 | 69.9 |
| ACT [20] | C | VGG | - | 65.7 | 69.5 |
| SMT [21] | C | I3D-CNN | K400 | 69.8 | 73.9 |
| MOC [14] | C | DLA34 | - | 70.8 | 78.0 |
| AVA [13] | C | I3D-VGG | - | 73.3 | 76.3 |
| STAD [22] | T | ViT-B | InternVid | 61.4 | 71.6 |
| WOO [5] | T | SF-R101 | K600 | 80.5 | 76.7* |
| TubeR [3] | T | I3D | IG+K400 | 80.7 | 81.3 |
| **Ours (MOGO)** | M | Mamba-B | K400 | **76.7** | **78.2** |

*: measured by [4].

We evaluate our MOGO model against SOTA methods on two datasets, with results summarized in Table 3. Our approach achieves an mAP of 76.7 on JHMDB while maintaining a low computational cost of 102-104 GFLOPs, leveraging an 8-frame Mamba-based backbone. Although our mAP is

slightly lower than top-performing Transformer-based methods such as WOO (80.5) and TubeR (80.7), these models rely on heavier architectures (e.g., ViT-B), which may increase computational overhead. In contrast, our model surpasses all CNN-based methods on JHMDB dataset, including I3D-CNN architectures like EESSK (64.4), and ACT (65.7), demonstrating the superior efficiency of Mamba-based modeling over conventional convolution designs. Compared to WOO (SFR101) and TubeR (CSN-152), our MOGO model reduces GFLOPs by approximately 60% and 57%, respectively, while only sacrificing 3.8–4.0 mAP points (from 80.5 to 76.7 for WOO, and 80.7 to 76.7 for TubeR) on the JHMDB dataset. Compared to TubeR (I3D), our MOGO model only sacrifices 3.1 mAP points on the UCF101-24 dataset. Part of this gap likely reflects the availability of strong Transformer pretraining (e.g., IG+K400) leveraged by TubeR and other Transformer-based methods, whereas comparable large-scale pretrained checkpoints for Mamba are still scarce.

**Comparison on GPU Memory Usage with EVAD [4].** We evaluate GPU memory consumption using the same settings as EVAD (an efficient video action detection method with significant performance improvements using Transformer), employing a single GPU with a batch size of 8. Both methods adopt token reduction strategy, where tokens are removed according to predefined rules. The results in Figure 5 demonstrate that: first, our method consistently consumes less GPU memory compared to EVAD. EVAD requires around 1.6 times more memory than our method, when token retention ratio is 0.4. Second, as the number of retained tokens increases, the GPU memory usage of EVAD grows significantly. In the extreme case where all tokens are retained, EVAD requires 2.8 times more memory than our method.

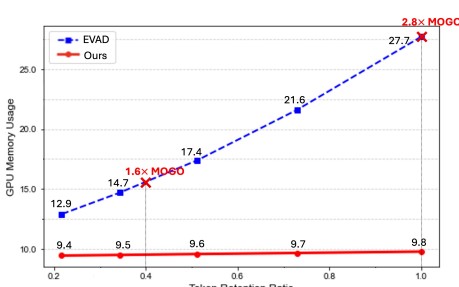

Figure 5: **Comparison of GPU memory usage between our method and EVAD under different token retention ratios.**

Therefore, MOGO maintains lower and stable GPU memory footprints compared to EVAD. **That means under the same hardware and experimental conditions, it can employ larger batch sizes to improve performance further.**

## 3.4 Ablation Studies

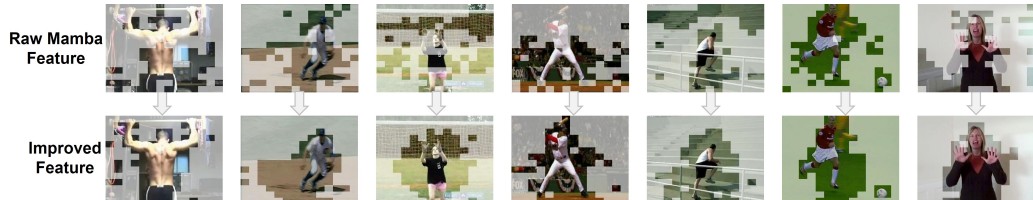

Figure 6: **Qualitative ablation: visualization of the impact of the token importance mechanism.** Row 1 shows raw Mamba outputs without the loss guidance. Row 2 shows outputs with our proposed importance modeling. Tokens inside target regions are more emphasized.

**Qualitative Ablation.** Unlike Transformers, which rely on attention mechanisms to compute token-to-token interactions, Mamba encodes temporal dependencies implicitly through sequential modeling. This makes it challenging to directly interpret token importance. As shown in the first row of Figure 6, raw Mamba outputs tend to produce scattered and less structured token representations. To address this, we propose a heuristic loss function (see Sec. 2.4), which encourages tokens inside the ground truth bounding boxes to receive higher importance. Furthermore, to avoid introducing significant computational overhead, we insert a lightweight MLP directly before the encoder tokens as shown in Table 1. This MLP outputs a token-wise importance score, which is then multiplied element-wise with the encoder tokens. With this, as shown in the second row of Figure 6, the token activations become more focused on semantically meaningful regions. More examples are provided in Figure S4.

**Quantitative Ablation.** The following ablation experiments are conducted on the JHMDB dataset. The findings are summarized in Table 4.

*Pretrained models.* Table 4(a) presents the results of ablation experiments conducted on pretrained models. All models employ a Mamba-based middle-sized model as the backbone, pretrained on

Table 4: **Quantitative ablation: experiments results.** (a) Pretrained models. Using different pretrained models as the encoder. MFT: mask with fine-tuning, MPT: mask with pretraining. (b) Encoder depth. enc.: encoder. (c) Decoder components. The EQ-Mamba and QVI-Mamba components are critical design elements. These components are evaluated by removing each one individually to assess their impact on the model's performance. dec.: decoder, md.: model, rmd.: removed. (d) Decoder input: temporal info. In the decoder, the top $k\%$ of tokens are selected and integrated with keyframe tokens. (e) Decoder input: keyframe info. Ex.: Exchange token positions of keyframe and temporal info. (f) Query number. (g) Ratio of total loss.

(a) Pretrained models.

| Model | Baseline | MFT | MPT |
|---|---|---|---|
| mAP | 51.7 | 65.0 | 31.2 |

(b) Encoder depth.

| $l_e$ | mAP | enc.GFLOPs (#param) |
|---|---|---|
| 24 | 60.4 | 76 (55.721M) |
| 32 | 66.2 | 101 (73.996M) |
| 40 | 61.1 | 126 (92.271M) |

(c) Decoder components.

| Case | dec.GFLOPs | md.GFLOPs |
|---|---|---|
| EQ rmd. | 2.098 | 103 |
| QVI rmd. | 0.466 | 102 |
| Proposed | 2.297 | 104 |

(d) Decoder input: temporal info.

| $k$ | mAP | dec.GFLOPs(#Param) | md.GFLOPs |
|---|---|---|---|
| 5 | 66.4 | 1.198 (7.228M) | 103 |
| 10 | 66.8 | 1.357 (7.228M) | 103 |
| 20 | 67.1 | 1.676 (7.228M) | 103 |
| 40 | 67.4 | 2.297 (7.228M) | 104 |
| 50 | 66.5 | 2.616 (7.228M) | 104 |
| 70 | 65.6 | 3.237 (7.228M) | 105 |

(e) Decoder input: keyframe info.

| Case | Removed | Ex. | Proposed |
|---|---|---|---|
| mAP | 53.7 | 53.9 | 69.0 |

(f) Query number.

| Case | 50 | 100 | 200 |
|---|---|---|---|
| mAP | 58.5 | 66.0 | 61.3 |

(g) Ratio of total loss.

| $\lambda$ | 0.1 | 0.5 | 1 | 2 |
|---|---|---|---|---|
| mAP | 73.3 | 75.6 | 74.4 | 74.4 |

the K400 dataset [23] with a resolution of 224×224. The baseline model achieves a mAP of 51.7. Introducing a mask with fine-tuning significantly boosts the mAP to 65.0. In contrast, using a mask with pretraining yields a decrease, achieving an mAP of 31.2. These results suggest that fine-tuning with masking is more effective than pretraining, likely due to better adaptation to the target task. Further discussion about pretrained models can be found in Section 4.

*Encoder depth.* The results in Table 4(b) indicate that using 32 layers in the encoder achieves the highest mAP of 66.2, with an encoder computational cost of 101 GFLOPs (73.996M parameters) and a model-level cost of 104 GFLOPs. Increasing the depth to 40 layers results in performance degradation (mAP 61.1) despite a higher computational cost of 126 GFLOPs (92.271M parameters), likely due to overfitting. Therefore, we set the encoder depth to 32 as the default.

*Decoder components.* Table 4 (c) investigates the decoder structure by ablating key components, EQ-Mamba (EQ) and QVI-Mamba (QVI), and comparing them against the proposed design. Removing EQ-Mamba or QVI-Mamba reduces the mAP significantly (possible theoretical reason is that removing EQ is equivalent to a random query directly entering Mamba, while removing QVI is equivalent to using inadequate video information), although decoder GFLOPs drops slightly. This demonstrates that every component of our design is indispensable but lightweight, further validating the correctness of our extrapolation from Transformer to Mamba as depicted in Figure 1.

*Decoder input: temporal information.* Table 4 (d) evaluates the impact of selecting the top $k\%$ of non-keyframe tokens in the decoder, which are integrated with keyframe tokens for processing. As $k$ increases from 5 to 70, the mAP im-

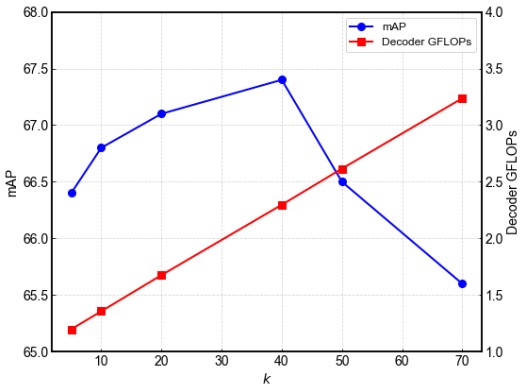

Figure 7: **The trends of mAP and decoder GFLOPs with varying numbers of retained non-keyframe tokens.** It demonstrates the importance of video feature tokens from non-keyframes as temporal information. We evaluate the impact of selecting the top $k\%$ of tokens.

proves from 66.4 to a peak of 67.4 at $k$=40, while decoder GFLOPs rise from 1.198 to 3.237, with the number of parameters remaining constant at 7.228M and model GFLOPs slightly increasing from 103 to 105. Beyond $k$=40, the mAP declines to 66.5 at $k$=50 and further to 65.6 at $k$=70, despite higher computational costs. This indicates that $k$=40 strikes an optimal balance between performance and efficiency, making it the most effective choice for this strategy. Figure 7 demonstrates the importance

of video information from frames other than the keyframe as temporal information. Intuitively, this drop when $k>40$ occurs because $k$ controls only the retention of non-keyframe tokens. We keep all keyframe tokens and perform detection on the keyframe, while non-keyframes serve primarily as contextual support. As $k$ increases, more low-utility background tokens from non-keyframes are admitted, which can dilute salient cues and slightly reduce performance.

*Decoder input: keyframe information.* Table 4 (e) compares different configurations of keyframe information usage. The one with keyframe information removed achieves an mAP of 53.7, while exchanging the token positions of the keyframe information and the temporal information (Ex.) slightly improves the mAP to 53.9. In contrast, the initial arrangement outperforms both alternatives with an mAP of 69.0. This performance gap demonstrates that the proposed design, leveraging the full potential of keyframe information, is the most effective approach for video token construction.

*Query number.* Table 4 (f) evaluates the effect of varying the number of queries on model performance. This limitation of using a fixed number of queries is discussed in Section 5.

*Ratio of total loss.* Due to our redesign of the loss function, the relationships between the different components have also changed, making the final results sensitive to the choice of weights. Therefore, a re-evaluation of the weight configurations is necessary to optimize performance. The findings are summarized in Table 4 (g), with parameters defined in Section 2.4. For the total loss ratio, adjusting $\lambda$ reveals that 0.5 is the optimal choice, achieving the highest mAP of 75.6.

## 4 Discussion

First, we discuss *innovation 1: pure Mamba-based architecture*:

**Extension to multi-label detection.** MOGO is originally designed for end-to-end detection, while AVA involves multi-label prediction per bounding box. To further evaluate adaptability, we modified the existing decoder to support multiple labels per bounding box without introducing an explicit classification branch. Corresponding adjustments were made to the loss and training setup. On the AVAv2.2 validation set, the adapted MOGO achieved an mAP of 16.2 (at step 26). This shows that MOGO can extend to multi-label settings with competitive accuracy. Training logs are presented in Figure S6. Another Mamba-based work [24] has reported their evaluations; however, a fair comparison would require additional details (e.g., architectural specifications and FLOPs calculation).

**Comparison with FlashAttention.** This comparison assesses how an optimized Transformer baseline performs relative to Mamba. We implemented a decoder variant that replaces the Mamba blocks with FlashAttention 2.8.2 (non-causal). The modified decoder reduces parameters from 7.228M to 5.321M and GFLOPs from 2.297 to 1.116, indicating higher computational efficiency. Trained on JHMDB with 4 × A40 GPUs for 50 epochs and a batch size of 32, it achieves an mAP of 70.1 (Initial loss: 7.7654, grad norm: 14.1749 → Final loss: 0.4765, grad norm: 1.8073). While FlashAttention offers higher efficiency, its mAP does not surpass our Mamba-based decoder. However, this motivates improving Mamba's low-level C++/CUDA kernels.

**Performance on longer video sequences and varying pretraining models.** We conducted experiments on longer video sequences using the UCF101-24 dataset. For each setup, we employed a corresponding pre-trained encoder from [7] and adjusted the decoder accordingly.

Table 5: **Performance on longer video sequences.**

| Ex. | Frames | Pretrained Model | Batch Size | mAP |
|-----|--------|------------------|------------|-------|
| 1 | 16 | K400 | 16 | 69.50 |
| 2 | 64 | Breakfast-actions-dataset | 4 | 41.90 |
| 3 | 64 | K400 | 4 | 63.43 |

All experiments were trained for 30 epochs on 2 NVIDIA A40 GPUs. The configurations and results are summarized in Table 5. These results indicate that our method maintains good performance on longer video sequences. Moreover, we observed that the choice of pre-trained model plays a crucial role: using K400 pretraining yields higher accuracy than using the Breakfast dataset [25]. To further investigate this factor, we trained with a new encoder pretrained on the SSV2 dataset [26] (2 A40 GPUs, 50 epochs, batch size 30) and obtained an f-mAP of only 64.1 on JHMDB (Initial loss: 7.9834, grad norm: 67.6360 → Final loss: 0.4847, grad norm: 6.6854). The results confirm that K400 pretraining remains superior to SSV2 in our setting.

**Training with long frames and autoregressive tracking.** Autoregressive trackers such as Track-Former [27] propagate track queries frame-by-frame to jointly detect and track objects across long videos without fixing a global clip length, illustrating an alternative way to scale beyond 64 frames. In contrast, our method currently relies on publicly available pretrained encoders [7], whose checkpoints

are trained for inputs up to 64 frames. Since action-detection mAP is highly sensitive to the encoder's pretraining quality and capacity, this limitation arises from data and model availability rather than the framework itself. In principle, our pipeline can process much longer sequences.

**Bidirectional scan in the encoder.** Our encoder employs a bidirectional scan. In Mamba, this means that at each position a forward state summarizes all preceding tokens and a backward state summarizes all following tokens; the token representation is then obtained by fusing these two states. As a result, each token can store richer information.

Next, we discuss *Innovation 2: video token construction mechanism*:

**Rethinking token selection: RLT-based and random token selection.** Inspired by Run-Length Tokenization (RLT) [28], we replace our original decoder-side token selection with two key components: (i) *token selection*, which prunes temporally redundant tokens while retaining tokens that change semantically across frames; and (ii) *run-length embedding*, which encodes how long a selected token remains stable and injects this temporal continuity into the token features before decoding. On JHMDB (4 GPUs, 60 epochs, batch size 32), this variant reaches an f-mAP of 72.1 (Initial loss: 8.4603, grad norm: $143.7739 \rightarrow$ Final loss: 0.4385, grad norm: 2.4390). While slightly below our best MOGO configuration, these results indicate that combining RLT-style token selection with run-length information is a promising direction for further improvement. The above is a heuristic attempt. For a stochastic baseline, we removed importance logits and randomly retained a fixed proportion of tokens per frame (40%), keeping all other settings identical to Figure 7. Trained for 30 epochs, this random-selection variant achieved an f-mAP of 67.2 (Initial loss: $7.9832 \rightarrow$ Final loss: 0.6982). Under the same environment and retention ratio, our importance-guided selection attains 67.4 mAP, indicating a consistent gain over random sampling.

**Generalizing the token-importance block to other methods.** To broaden applicability beyond our own architecture, we integrate the token-importance module into TubeR [3] and evaluate on JHMDB, comparing the standard full-token setting with variants that retain only a subset of non-keyframe tokens. Using the released JHMDB-pretrained CSN-152 checkpoint, the detector attains 71.2 mAP after one epoch. We therefore adopt the pretrained setting as the full-token baseline. With our importance scores guiding token reduction, keeping 80% of non-keyframe tokens yields 72.05 mAP, and keeping 40% yields 72.0 mAP. Note that while the Transformer in TubeR inherently possesses an attention-based token importance mechanism, we align its setup with our proposed method by using the outputs of the trainable MLP to represent token importance within the encoder. The intermediate frames are also designated as key frames. For the loss, however, we adhere to the original TubeR implementation. These results are on par with the full-token baseline in small-scale experiments.

**Keyframe-centric detection without an action-switch head.** Our approach does not employ an explicit *action-switch head*; instead, it follows a DETR-style formulation with Hungarian matching over class logits and bounding box predictions. Conceptually, the method can be viewed as image-style action detection: we predict on a designated keyframe (typically the middle frame) while importing **contextual tokens** from preceding and following frames. This design helps the model sense how a person's action shifts across time, but it does not explicitly model precise action boundaries or subtle temporal transitions.

## 5 Conclusion and Future Work

Our proposed MOGO framework introduces a pure Mamba-based architecture for end-to-end video action detection, eliminating reliance on Transformer components. This design addresses limitations noted in prior work, such as the complexity overhead in [4], by leveraging Mamba's state-space modeling for efficient processing of video sequences. This results in a significant reduction in computation while maintaining competitive precision. Additionally, our video token construction mechanism can obtain important token information across spatiotemporal frames. In this way, important cues (closely related to the action prediction) can be effectively retained.

MOGO's performance is among the top-tier Transformer-based methods, though it has not surpassed the very best ones. This limitation stems from two factors: first, GFLOPs reduction may trade off some precision; second, the scarcity of pre-trained Mamba models limits feature representation quality, unlike the extensive pretraining pools available for Transformers. A further limitation is the use of a fixed query; in its current form, the model cannot reliably handle scenes with more than 100 people. However, this study highlights Mamba's promise in this new field. Future work will focus on developing richer pretrained models to help enhance performance.

## Acknowledgments and Disclosure of Funding

**Funding:** This work received no third-party funding or support. **Competing Interests:** The authors declare no competing interests.

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

# A Technical Appendices and Supplementary Material

## A.1 Why Mamba, not Transformers?

We argue that the proposed Mamba-based framework is well-suited for video action detection due to the following reasons:

**Linear Complexity.** Transformers rely on self-attention to model long-range dependencies, resulting in quadratic complexity with respect to sequence length. Specifically, for a sequence of length $L$ and feature dimension $d$, the computational cost of self-attention is:

$$\text{Cost}_{\text{Transformer}} = \mathcal{O}(L^2 d). \tag{S1}$$

In contrast, the Mamba block updates hidden states via a linear state space model:

$$\text{Cost}_{\text{Mamba}} = \mathcal{O}(L\,d^2) \approx \mathcal{O}(L). \tag{S2}$$

This is critical for video sequences in real-time or resource-constrained settings.

**Causal Modeling.** Equation (1) models causal dependencies. Each output depends only on the previous hidden state and current input. This aligns well with the nature of action detection, which relies on the context of preceding and current frames. (However, Mamba may not be well-suited for non-causal tasks such as image restoration, which require access to future tokens.) By contrast, general Transformer-based models perform undifferentiated attention over a large number of previous, current and **future** information tokens, which can introduce redundancy and noise. Mamba, therefore, offers a more efficient and focused modeling paradigm for video-based tasks. This is essentially a coarse-to-fine design. While Transformer attention is detail-oriented (attention), Mamba views the input-output process as a holistic dynamic system. This allows us to design global optimization strategies. From a theoretical perspective, it supports more principled modeling than heuristically stacking attention layers.

**However, Mamba lacks explicit attention mechanism.** This is why we introduce a minimalistic linear projection (MLP) layer coupled with a purposefully designed loss function to simulate attention-like behavior. This enhancement enables the model to focus on important tokens while preserving the framework performance.

## A.2 Related Works

**Transformer-Based Action Detection.** Recent Transformer-based methods often employ a two-stage pipeline, combining 2D backbones for actor localization with 3D backbones for temporal context extraction. For instance, CycleACR [29] uses a Transformer to model actor-context relations, enhancing detection through cyclic consistency across frames. Query-based approaches like TubeR [3] and WOO [5] build on DETR [2], predicting action tubes and categories with the quadratic complexity of attention mechanisms. In VAT [30], spatiotemporal features are aggregated around actors. Despite these advances, the computational burden of Transformers remains a challenge, especially for long sequences, motivating exploration of efficient alternatives.

**Mamba-Related Applications.** The Mamba architecture [6], with its linear-time state-space modeling, has emerged as a lightweight alternative to Transformers. VideoMamba [7] adapts Mamba for video classification, achieving competitive top-1 accuracy on K400 by leveraging selective state-space mechanisms to model temporal dependencies efficiently. VideoMambaPro [31] further enhances this by addressing historical decay and element contradiction. MS-Temba [8] extends Mamba to multi-scale temporal localization on datasets like MultiTHUMOS. However, while Mamba excels in classification and localization, its application to video action detection remains underexplored, providing a gap that our MOGO framework addresses with a pure Mamba-based end-to-end solution.

**Hybrid Approaches**. Mamba has also been explored in hybrid models like MambaVision [32], a work combining Mamba and Transformer for vision tasks, and Simba [33], which augments Mamba with graph networks for skeletal action recognition. Recently, TransMamba [34] introduced a hybrid Transformer-Mamba backbone that adapts attention mechanisms for faster inference while preserving detection accuracy. Similarly, MV-GMN [35] combines rule-based and KNN-based methods with state-space models to enhance robust action recognition.

## A.3 Implementation Details

**Data Processing.** We decode raw videos using the Decord backend and uniformly sample 8 frames for training and evaluation. The input frames are resized and cropped to 224×224 resolution. To construct training and testing samples, we rely on the provided split lists from relative datasets, and retain only clips that have valid annotation files. The underrepresented classes were balanced. For data augmentation during training, we include random short-side scale jittering ([256, 320]), random horizontal flipping, and optional color jittering (PCA-based lighting augmentation). During testing, center cropping is applied after resizing the shorter side to 256. All frames are normalized using the mean and standard deviation values, respectively. Bounding boxes are clipped to image boundaries to avoid numerical instability. Per-frame annotations are matched using the keyframe index of each sampled clip, and annotations are encoded as bounding boxes and class indices for downstream use.

**Training Engine.** All experiments are conducted using PyTorch with mixed-precision training (torch.cuda.amp.autocast). Gradient clipping is applied to stabilize training. Both the learning rate and weight decay are scheduled using precomputed cosine decay curves and updated at every step. We adopt the AdamW optimizer with betas set to (0.9, 0.999), a weight decay of 0.05, and an initial learning rate of 1e-4. The learning rate is scaled according to the effective batch size and follows a cosine annealing schedule. Training is performed for 50 epochs, with a 5-epoch warmup phase. Since the designed model has an advantage in GPU memory usage, we recommend using a larger batch size for training ($\geq$30). We use a batch size of 30 per GPU and set the update frequency to 1. Training is distributed using torch.distributed, with full synchronization across processes. The total loss consists of a standard object detection loss, including classification, bounding box regression, and GIoU terms, along with a token importance loss. The latter encourages alignment between the model-predicted token importance and the ground-truth spatial token relevance derived from bounding box annotations, computed using binary cross-entropy. (Specifically, we first generate normalized spatial coordinates for all tokens in each frame. Given $H \times W$ spatial patches and $T$ temporal frames, this produces a tensor of shape $[T \times HW, 2]$. These are compared with all bounding boxes to obtain a binary inclusion map.) Evaluation is done using torchmetrics with COCO-style AP. For each sample, we extract predicted logits and bounding boxes, apply softmax to logits, and select the top score per query. Each training batch consists of $B$ video clips, where each clip is represented as a tensor of shape $[3, T, H, W]$. Annotations are processed into lists of dicts with keys *boxes* and *labels*, compatible with TorchMetrics and detection criteria. We log training loss, learning rate, gradient norm, and validation mAP at each step using a custom logger. We use torchrun to launch distributed training with synchronized logging and gradient updates across GPUs.

**Model.** The core architecture consists of a Mamba-based spatiotemporal encoder and a lightweight decoder. The video encoder is designed based on the Mamba sequence modeling block. Following standard video modeling practices, we adopt a 3D convolutional layer to embed $T$-frame video clips into patch tokens of shape $[B, T \times N, D]$, where $N$ is the number of spatial patches per frame. Each patch is then projected using a temporal tubelet embedding (kernel size = 1). Spatial and temporal positional embeddings are added separately. The token features are further enhanced by a stack of $l_e$ Mamba-based encoder blocks, each composed of a selective copy mechanism and RMSNorm. To improve efficiency and enable token reduction, we introduce a simple yet effective module that estimates the importance of each token via an MLP. This module predicts a scalar score for each token, and soft masks are applied before the encoder layers. For the decoder, a learnable query embedding of shape $[100, D]$ is provided as input, and an $l_d$-layer Mamba decoder processes over the encoded memory tokens. The final outputs are passed through a classification head and a bounding box regression head (MLP with sigmoid activation). We adopt RMSNorm as the default normalization layer. The encoder is initialized from a pretrained checkpoint [7] on a large-scale masked video modeling task.

## A.4 Ablation Studies - Supplementary

Table S1: **Quantitative ablation (supplementary): experiments results.** (a) Decoder depth. (b) Weights of detection loss. $w = w_{\text{cls}}: w_{\text{box}}: w_{\text{ovl}}$

<table>
<tr><td colspan="4">(a) Decoder depth.</td><td colspan="6">(b) Weights of detection loss.</td></tr>
<tr><td>$l_d$</td><td>1</td><td>3</td><td>6</td><td>$w$</td><td>11:1:1</td><td>5:1:1</td><td>1:1:1</td><td>1:5:2</td><td>2:3:2</td></tr>
<tr><td>mAP</td><td>75.6</td><td>71.7</td><td>32.0</td><td>mAP</td><td>67.2</td><td>68.6</td><td>46.3</td><td>4.2</td><td>35.6</td></tr>
</table>

*Decoder depth.* Table S1(a) shows that a 1-layer decoder gives the best mAP (75.6), while 3 or 6 layers degrade performance, likely due to overfitting. We adopt a single-layer decoder by default. Therefore, MOGO (Mamba Only Glances Once), i.e., our Mamba-based decoder that *glances only once*. For the corresponding training dynamics, please refer to Figure S1.

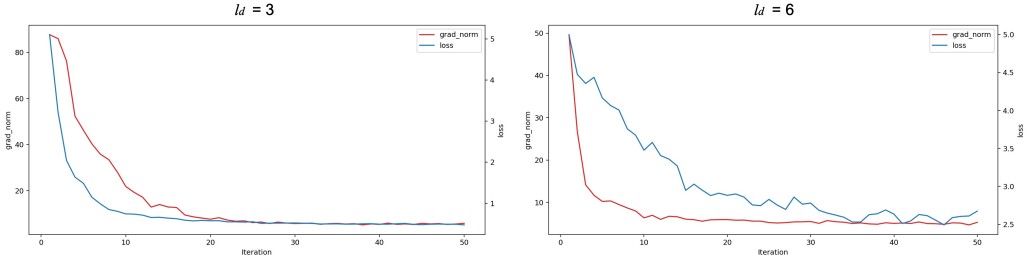

Figure S1: **Training curves and diagnostics.** Training loss (right axis) and gradient norm (left axis) for decoder depths $l_d > 1$.

*Weights of detection loss.* This set of experiments was conducted with parameter $\lambda$ (Table 4) fixed. For the detection loss (Table S1(b)), emphasizes the classification weight ($w_{cls}$) proves beneficial, with ratios of 5:1:1 and 11:1:1 yielding strong mAP scores of 68.6 and 67.2, respectively, compared to lower performance at other settings (e.g., 46.3 at 1:1:1). This suggests that a larger $w_{cls}$ enhances detection accuracy. We note that the 1:5:2 configuration performs poorly; controlled re-runs confirm this trend. When using 1:5:2 weights, we observed high loss (epoch[0]:6.0x/5.9x → epoch[29]:2.x) and unstable grad norms (400 after 20 epochs), while mAP stayed under 0.05. In contrast, the 5:1:1 configuration yields stable optimization dynamics: the loss decreases steadily (epoch[0]:8.0x → epoch[29]:0.8x), gradient norms consistently drop after 10 epochs, and the mAP rises above 0.6. So now we have enough evidence to say that the 1:5:2 setting may overwhelm optimization with box/ovl weights, preventing the model from learning effective action features.

## A.5 Video-level Results

**Video-Level Detection mAP.** We provide the video-mAP results on JHMDB and UCF101-24 datasets. As shown in Table S2, Table S3 and Table S4, our method achieves lower video-level mAP compared to WOO while outperforming both WOO and TubeR in terms of model size, GFLOPs, and throughput. Because our method detects actions on a designated keyframe within each clip (e.g., the middle frame in an 8-frame snippet), using the remaining frames as temporal context, frame-level metrics such as f-mAP@0.5 are a more appropriate measure. In contrast, video-level mAP emphasizes constructing continuous tubes over the entire clip, which may understate the strengths of our keyframe formulation.

Table S2: **Per-class AP on JHMDB under video-level detection evaluation.** (IoU: 0.2)

| Class | brush_hair | catch | clap | climb_stairs | golf | jump | kick_ball | pick | pour | pullup | push |
|-------|-----------|-------|------|--------------|------|------|-----------|------|------|--------|------|
| AP | 0.833 | 0.553 | 0.743 | 0.725 | 1.000 | 0.437 | 0.791 | 0.843 | 1.000 | 0.937 | 0.976 |

| Class | run | shoot_ball | shoot_bow | shoot_gun | sit | stand | swing_baseball | throw | walk | wave |
|-------|-----|-----------|-----------|-----------|-----|-------|----------------|-------|------|------|
| AP | 0.486 | 0.450 | 1.000 | 0.870 | 0.297 | 0.421 | 0.667 | 0.181 | 0.576 | 0.465 |

Table S3: **Per-class AP on UCF101-24 under video-level detection evaluation.** (IoU: 0.2)

| Class | Basketball | BasketballDunk | Biking | CliffDiving | CricketBowling | Diving | Fencing | FloorGymnastics |
|-------|-----------|----------------|--------|-------------|----------------|--------|---------|-----------------|
| AP | 0.556 | 0.519 | 0.596 | 0.757 | 0.404 | 0.974 | 0.767 | 0.981 |

| Class | GolfSwing | HorseRiding | IceDancing | LongJump | PoleVault | RopeClimbing | SalsaSpin | SkateBoarding |
|-------|-----------|-------------|------------|----------|-----------|--------------|-----------|---------------|
| AP | 0.886 | 0.940 | 0.227 | 0.864 | 0.928 | 0.884 | 0.447 | 0.993 |

| Class | Skiing | Skijet | SoccerJuggling | Surfing | TennisSwing | TrampolineJumping | VolleyballSpiking | WalkingWithDog |
|-------|--------|--------|----------------|---------|-------------|-------------------|-------------------|----------------|
| AP | 1.000 | 0.913 | 0.806 | 0.596 | 0.494 | 0.448 | 0.189 | 0.817 |

Table S4: **Comparison of video-level detection performance and model efficiency.**

| Method | Video-mAP@0.2 (U) | Video-mAP@0.2 (J) | Video-mAP@0.5 (U) | Video-mAP@0.5 (J) | #Params | GFLOPs | Throughput |
|--------|-------------------|-------------------|-------------------|-------------------|---------|--------|------------|
| WOO [5] | 74.4 | 70.0 | 55.8 | 69.5 | 314M (head) | 252–378 | 147–176 |
| TubeR (I3D) [3] | **85.3** | **81.8** | **60.2** | **80.7** | - | 240 | 64 |
| Ours | 70.8 | 67.9 | 39.4 | 42.0 | **82M** | **102–104** | **256** |

**Video-Level Classification mAP.** We further provide the video-level classification mAP results on the JHMDB and UCF101-24 datasets. We conduct this experiment because the dataset provides video-level category labels. Therefore, we aim to evaluate the model's ability to extend frame-level predictions. Our method follows a sliding-window inference strategy to compute video-level predictions. Given an input video, we first extract RGB frames and apply standard preprocessing including resizing, normalization, and batching. The video is divided into clips, and each clip is fed into the model to obtain per-query logits. We compute the average of softmaxed logits (excluding the background class) across all queries and clips. Finally, the class probabilities are aggregated over the video to produce the final prediction. The performance is measured by per-class AP and the overall mean AP. On the JHMDB dataset, as shown in Table S5, our model achieves strong results across most categories, particularly excelling in actions such as *golf*, *pullup*, *push*, and *shoot_bow*. Overall, our approach attains a video-level mAP of 0.812, demonstrating robust performance on this benchmark.

Table S5: **Per-class AP on JHMDB under video-level classification evaluation**.

| Class | brush_hair | catch | clap | climb_stairs | golf | jump | kick_ball | pick | pour | pullup | push |
|---|---|---|---|---|---|---|---|---|---|---|---|
| AP | 0.932 | 0.711 | 0.908 | 0.981 | 1.000 | 0.811 | 0.986 | 0.870 | 1.000 | 1.000 | 1.000 |

| Class | run | shoot_ball | shoot_bow | shoot_gun | sit | stand | swing_baseball | throw | walk | wave | |
|---|---|---|---|---|---|---|---|---|---|---|---|
| AP | 0.560 | 0.555 | 1.000 | 0.951 | 0.378 | 0.486 | 0.996 | 0.560 | 0.751 | 0.616 | |

On the UCF101-24 dataset, it achieves an mAP of 0.971. This result demonstrates that our model possesses strong classification capabilities at the video level without considering bounding box predictions.

Table S6: **Per-class AP on UCF101-24 under video-level classification evaluation**.

| Class | Basketball | BasketballDunk | Biking | CliffDiving | CricketBowling | Diving | Fencing | FloorGymnastics |
|---|---|---|---|---|---|---|---|---|
| AP | 0.927 | 1.000 | 0.999 | 0.993 | 0.902 | 1.000 | 1.000 | 0.861 |

| Class | GolfSwing | HorseRiding | IceDancing | LongJump | PoleVault | RopeClimbing | SalsaSpin | SkateBoarding |
|---|---|---|---|---|---|---|---|---|
| AP | 0.929 | 1.000 | 1.000 | 0.993 | 1.000 | 1.000 | 1.000 | 0.972 |

| Class | Skiing | Skijet | SoccerJuggling | Surfing | TennisSwing | TrampolineJumping | VolleyballSpiking | WalkingWithDog |
|---|---|---|---|---|---|---|---|---|
| AP | 0.980 | 0.924 | 0.851 | 1.000 | 0.981 | 1.000 | 0.996 | 0.996 |

## A.6 Result Analysis and Visualization

**Per-Class Metric Visualization.** To further investigate the importance of keyframe information in our framework, we conduct a study comparing three variants: (1) *Ours*, the full model using both selected tokens and the fixed keyframe segment; (2) *Exchange*, where the keyframe segment is exchanged with non-keyframe tokens; and (3) *Remove*, where the keyframe segment is discarded altogether. As shown in Figure S2, our method consistently outperforms the baselines across most action classes, indicating that preserving keyframe information is crucial for accurate action localization. In particular, categories such as *golf*, *shoot_bow*, and *pullup* benefit significantly from retaining discriminative keyframe cues.

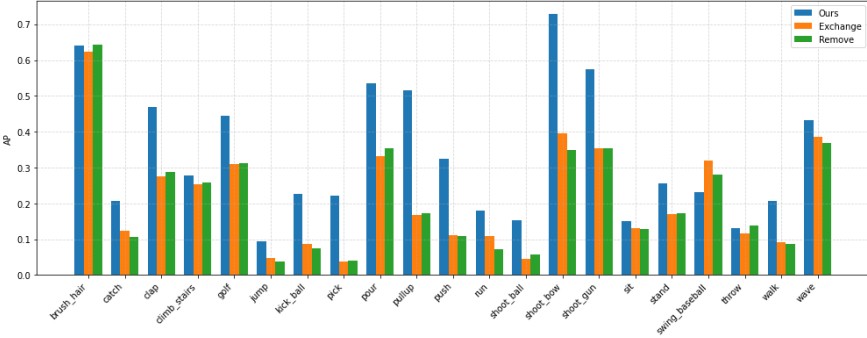

Figure S2: **Per-class AP(0.5:0.95) under different keyframe strategies on the JHMDB dataset.** *Ours* uses both the preserved keyframe and selected tokens. *Exchange* swaps keyframe tokens with other temporal tokens. *Remove* drops keyframe information completely. Preserving keyframe cues leads to the best performance across most categories.

Notably, the impact of certain parameters on mAP may vary slightly across datasets. We ablate the matcher's coefficients to understand the role of classification versus localization in action detection on the UCF101-24 dataset. As shown in Figure S3, using balanced weights (1:1:1) generally yields higher AP across a majority of the 24 classes. Emphasizing classification via (3:1:1) slightly improves a few classes (e.g., *Skiing*).

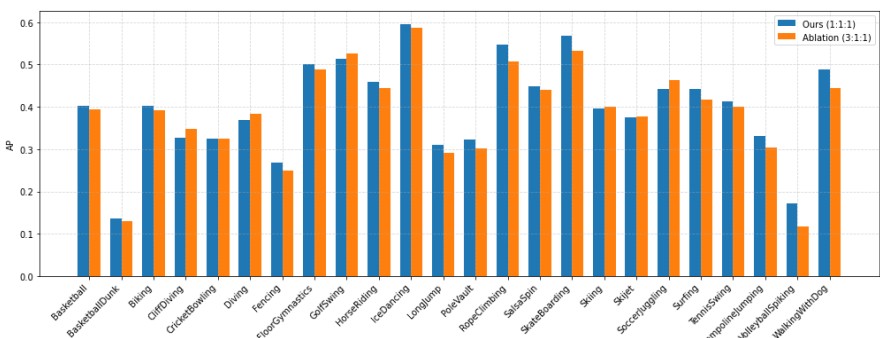

Figure S3: **Per-class AP(0.5:0.95) under different matcher weight ratios on the UCF101-24 dataset.** We compare the decoder performance using the default matcher weights (1:1:1) versus a biased setting (3:1:1).

**Learning Token Importance without Explicit Attention.** Unlike Transformers, which rely on explicit attention mechanisms to compute token-to-token interactions, Mamba encodes temporal dependencies implicitly through sequential modeling. This makes it challenging to directly interpret token importance. However, token selection is part of our design. As shown in the first rows of Figure S4, raw Mamba outputs tend to produce scattered and less structured token representations. To address this, we propose a heuristic loss function (see Section 2.4), which encourages tokens inside the ground truth bounding boxes to receive higher importance. Furthermore, to avoid introducing significant computational overhead, we insert a lightweight MLP directly before the encoder tokens as shown in Table 1. This MLP outputs a token-wise importance score, which is then multiplied element-wise with the encoder tokens. Notably, the output dimension of the MLP matches the number of encoder tokens, enabling one-to-one correspondence. With this, as shown in the second row, the token activations become more focused on semantically meaningful regions.

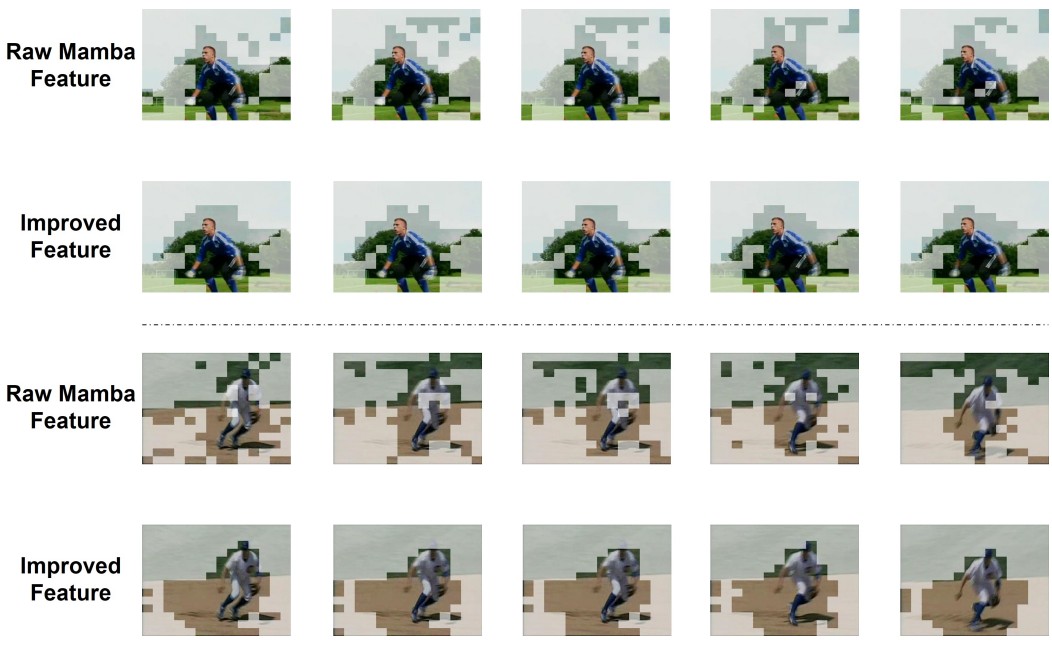

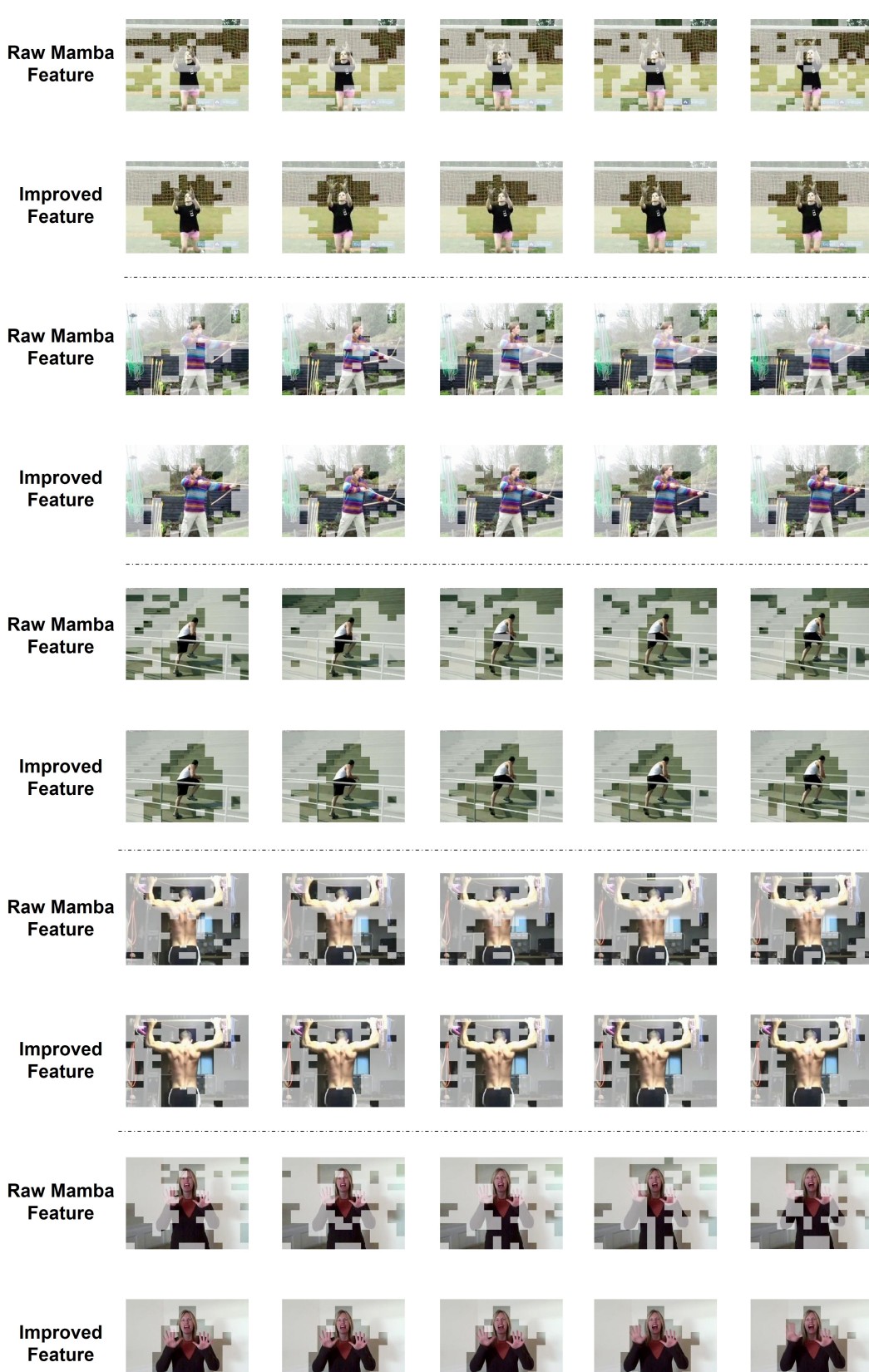

Figure S4: **Visualization of token importance.** Row 1 shows raw Mamba outputs without the loss guidance. Row 2 shows outputs with our proposed importance modeling. Tokens inside target regions are more emphasized.

**Spatial Distribution of Preserved Tokens and Corresponding Prediction Results.** To better understand the effectiveness of our token selection mechanism and the final prediction results, we visualize the spatial distribution of selected tokens alongside the predicted bounding boxes for some classes as shown in Figure S5. It is evident that the retained tokens are highly concentrated around target subjects and action-relevant regions, showing strong alignment with the predicted bounding boxes. In particular, actions such as *clap* involve fine-grained upper-body motions, where the preserved tokens closely overlap with the hands and head regions. This confirms that our token importance module effectively captures discriminative cues, helping the decoder focus on relevant spatial areas and improving localization precision.

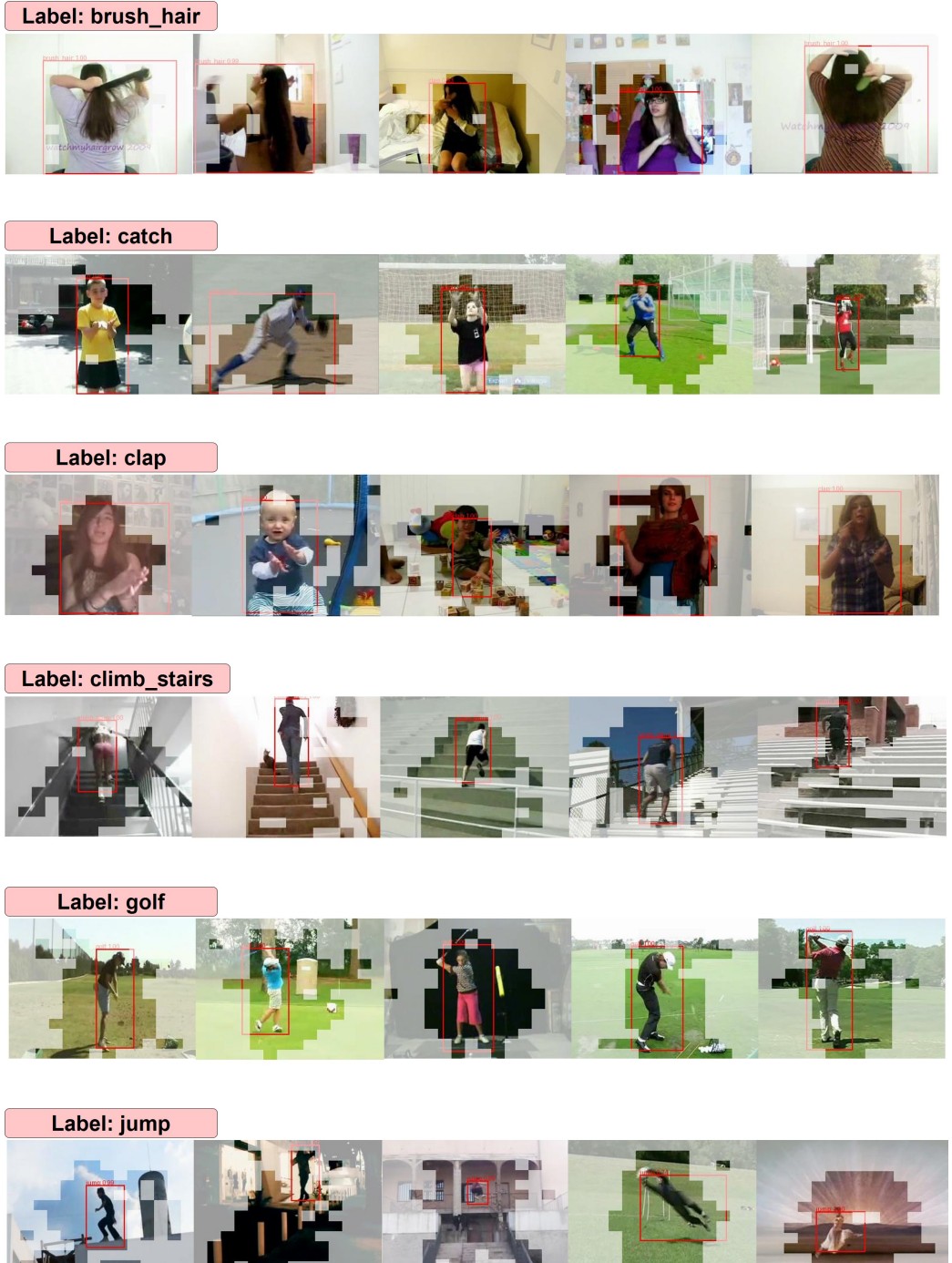

**Label: kick_ball**

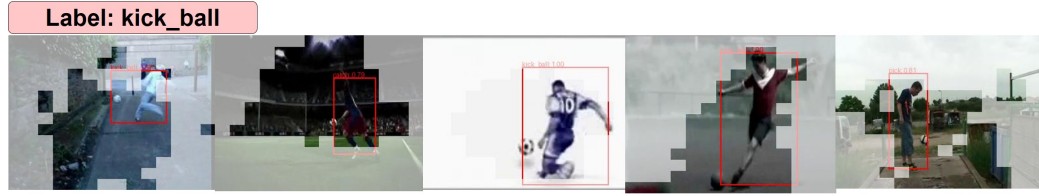

**Label: pick**

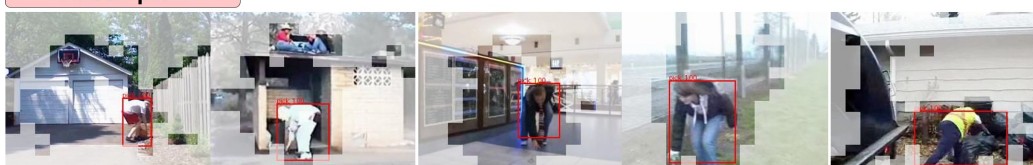

**Label: pullup**

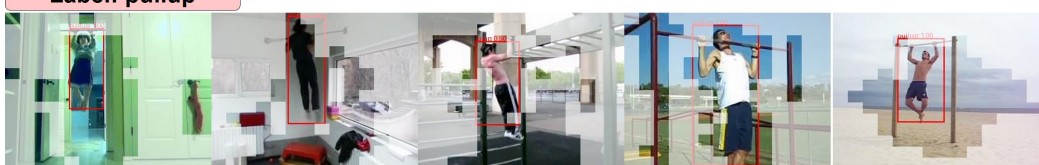

**Label: push**

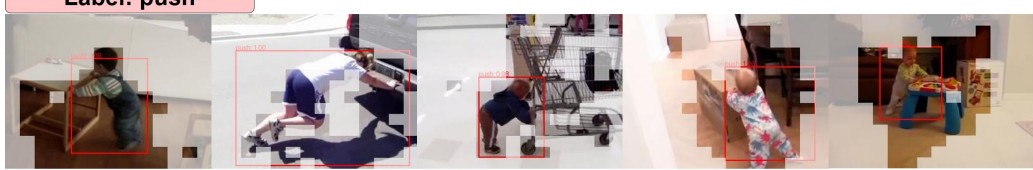

**Label: run**

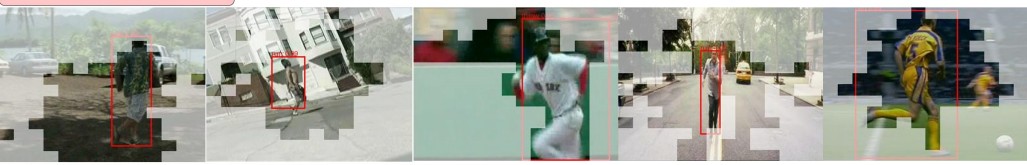

**Label: shoot_ball**

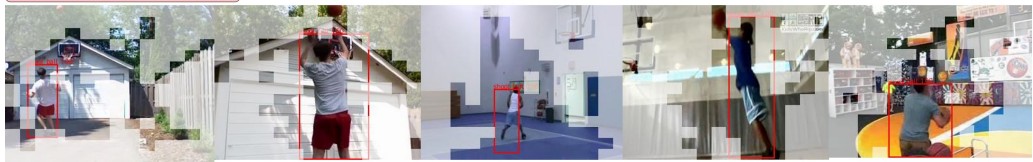

**Label: sit**

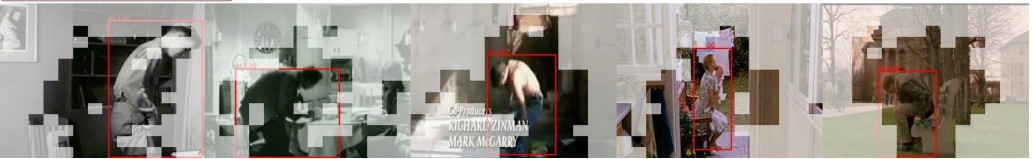

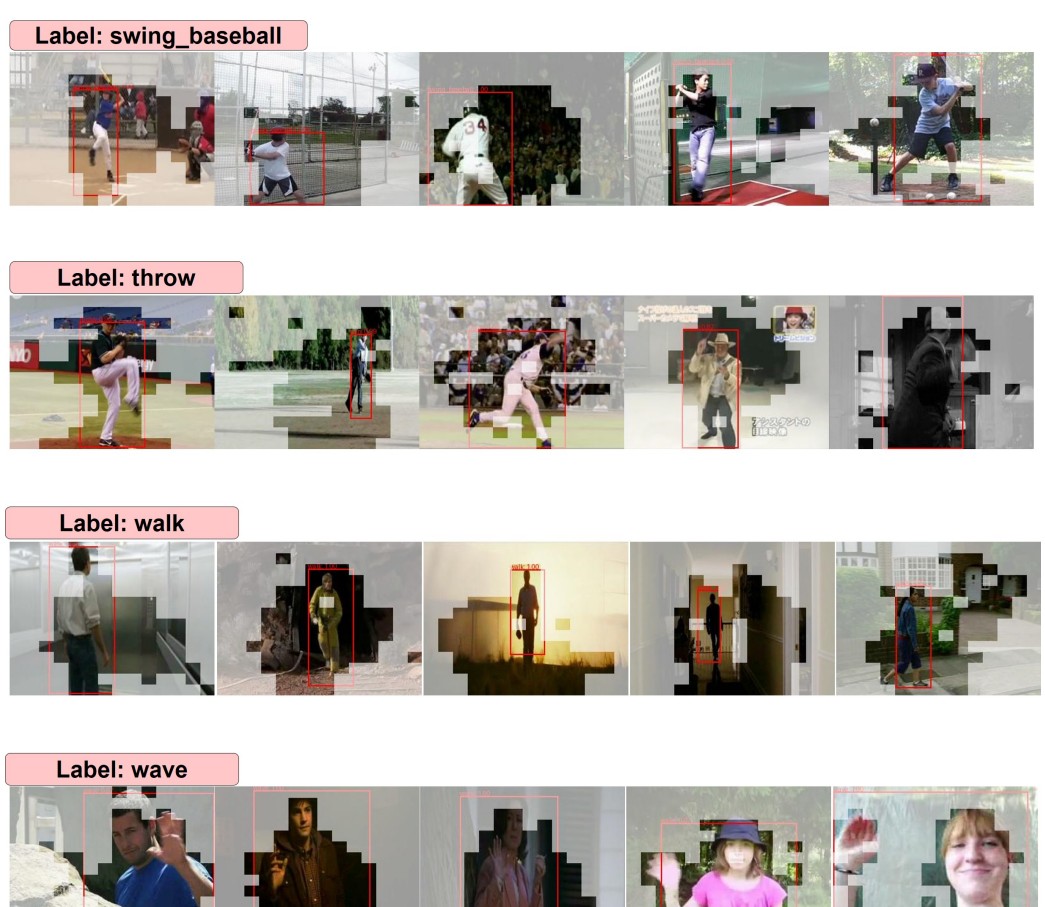

Figure S5: **Visualization of preserved tokens by token importance block on the JHMDB dataset.** Important cues such as people can be effectively retained and are closely related to the bbox prediction of the decoder.

**Extension to Multi-Label Detection.** On the AVA v2.2 validation set, the adapted MOGO achieved an mAP of 16.2 (at step 26). This demonstrates that MOGO can flexibly extend to multi-label settings with competitive accuracy. Training logs are presented in Figure S6.

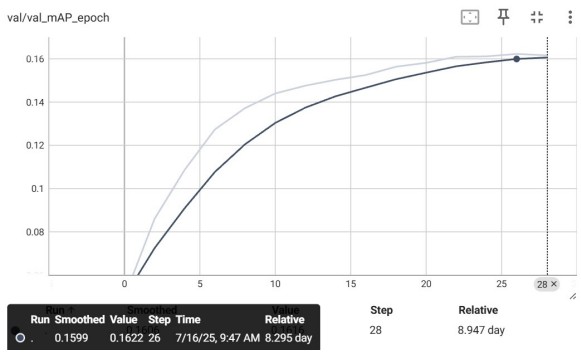

Figure S6: **Training logs on the AVA dataset.**

## A.7 Architecture Algorithm

To further clarify our architectural design, we present a step-by-step breakdown of the key modules in the form of pseudocode. Our approach introduces a token importance mechanism into video understanding pipelines, enabling dynamic token weighting and selection for efficient and effective representation learning.

---

**Algorithm 1** Token Importance-Guided Frame Encoding

---

**Require:** Video input $x \in \mathbb{R}^{B \times C \times T \times H \times W}$
**Ensure:** Feature representation $f$, importance logits $s$
 1: $x \leftarrow \text{PatchEmbed}(x)$ {Convert frames to patch tokens}
 2: Add spatial positional embedding to each frame
 3: Add temporal positional embedding across frames
 4: $s \leftarrow \text{ImportanceMLP}(x)$ {Predict token importance scores}
 5: $x \leftarrow x \odot \sigma(s)$ {Weight tokens by importance (sigmoid)}
 6: $f \leftarrow \text{MambaEncoder}(x)$
 7: **return** $f, s$

---

**Algorithm 2** Top-$k$ Token Selection per Frame

---

**Require:** Feature $f \in \mathbb{R}^{B \times N \times D}$, importance $s \in \mathbb{R}^{B \times N}$
**Ensure:** Reduced feature $f_{\text{selected}}$
 1: Divide $f$ into $T$ temporal segments, each with $N_T = N/T$ tokens
 2: **for** each frame $t = 1$ to $T$ **do**
 3:     $s_t \leftarrow s[:, t \cdot N_T : (t+1) \cdot N_T]$
 4:     $f_t \leftarrow f[:, t \cdot N_T : (t+1) \cdot N_T, :]$
 5:     Select top-$k$ indices by $s_t$
 6:     $f_t' \leftarrow \text{Gather } f_t$ using top-$k$ indices
 7:     Append $f_t'$ to $f_{\text{selected}}$
 8: **end for**
 9: **return** $f_{\text{selected}}$

---

**Algorithm 3** Object Detection with Mamba Decoder

---

**Require:** Memory $f_{\text{memory}}$, learnable queries $q \in \mathbb{R}^{Q \times D}$
**Ensure:** Predicted boxes $\hat{b}$, class logits $\hat{y}$
 1: $q \leftarrow \text{Repeat } q$ for each batch and initialize $tgt$
 2: $hs \leftarrow \text{MambaDecoder}(tgt, f_{\text{memory}})$
 3: $\hat{y} \leftarrow \text{ClassHead}(hs)$ {Predict object class}
 4: $\hat{b} \leftarrow \text{BoxHead}(hs)$ {Predict bounding box (normalized)}
 5: **return** $\hat{y}, \hat{b}$

---

Together, these components define the core workflow of our Mamba-based action detection framework.

