# OpenReview forum: "Mamba Only Glances Once (MOGO): A Lightweight Framework for Efficient Video Action Detection"
_NeurIPS.cc/2025/Conference — NeurIPS 2025 poster_

### Official Review · Reviewer_Pwyx · 2025-06-22

**Clarity:** 2
**Significance:** 3
**Originality:** 2
**Rating:** 5
**Confidence:** 5

**Summary:**

this paper propses a mamba based backbone for efficient spatio-temporal video action detection and a novel token-selection scheme to reduce compute while working on video data.

**Questions:**

Please see the weakness section.
- How many layers of encoder do you use? How many mamba layers are being used?

**Ethical Concerns:**

["NO or VERY MINOR ethics concerns only"]

**Final Justification:**

Dear Authors:

Thank you so much for your detailed rebuttals, additional experiments. I have increased my score accordingly. I would love to see your paper at NeurIPS, and am excited by how this might open up an area for linear-time spatio-temporal action detection.

However, please note, as per **what the respected ACs mentioned**, i am not sure what constitutes as ```major revisions.```

I would suggest categorizing your (promised) changes in a GLOBAL comment as:
(1) Changes you have made which are just clarifications on the experiments **already** present in the paper.
(2) Changes which are **brand new**, , and may be considered as ```substancial``` changes,  for eg, adding rle, ava experiments, discussions on table 14 of video-mamba paper not being reliable, flash attention baseline, discussions on training for >64 frames aka trackformer.


To be fair in the review process, the ```ultimate decision lies with the respected AC, PC's who know far more```, and whether they will ```allow``` such substancial revisions,

Thanks for your understanding,

reviewer

**Limitations:**

Addressed.

**Quality:**

2

**Strengths And Weaknesses:**

i am glad this paper landed in my lap , i haven't read a lot of papers in spatio-temporal action detection for a while, and **this is a really refreshing attempt** to see people still working in this area.

the question i will leave to discretion of other reviewers is:  **does application of mamba to this domain count as novel**. since mamba is already known, and its application to spatio-temporal action detection is not known, i will count this as a novel work . **my bet is a lot of people will think it is not novel**. but i feel that this work is something that will **greatly benefit** our community, and **open up an area for efficient inference in spatio-temporal video action detection**, and i am excited to see where it goes.

strengths:
- to the best of my knowledge, i am not aware of a mamba based architecture in spatio-temporal video action detection , this makes the sequence modelling linear as a function of sequence length, as opposed to transformers. **There is video-mamba paper**, which does it in general video understanding, but not in this area.

- halving the flops (102-104) over 240 flops in josies paper (tuber) seems pretty good.
- the token importance block seems to be independent of the length of video being processed (table 1). This seems good for efficiency.


weaknesses

- how many encoder blocks you use. If it's more than 1, then mamba glances again and again, and i dont know how it justifies the title , mamba glances only once? is it because you only apply token selection once (at the beginning before encoding)?

- line 31: mamba is a good concept, but the performance in table 4 still lacks behind tuber. **Can u plz plz train MOGO on IG+ K400 :-), and not just K400**, that should help close the gap with TubeR.

- i think one baseline should be against flash attention v3.  even though it is not linear, but it would be interesting to see how optimized transformer  fares against mamba.

- where are the results on ava 1.1, ava 2.2? josie showed those when she wrote her TubeR paper, which the authors compared against. so those experiments should also be shown.

- where are the vmap metrics? reporting those at 0.5 iou threshold is a standard practice in literature. table 7 in supp only shows vmap @0.2. it needs more numbers at 0.5 iou also.

- in table 1, decoder takes 100 queries. what if there are more than 100 people in the scene? what if we dont know the number of people to detect for action detection before hand. the network wont work in that case. ucf/jhmdb dont have those cases, but that doesnt make the paper bulletproof. Please add this in limitation.

- token selection blocks assumes a keyframe, and selects top k tokens per frame. This assumes that “equal number” of k tokens per frame are “semantically important”. i guess a better would be to use rle compression over time. check this paper, it should help you guys out :-) [1], it was neurips spotlight last year i think.  the token selection block seems too simple in my opinion, and if authors can make it a bit better, it would be great [1].

- **formatting suggestions**: the paper seems to sparse, too much white space, and figures dont look pleasing to the eye. while the official guidelines dont penalize authors on their language, and how they wrote the paper, my suggestion would be to shift all the figures to the right, and bring some content from the supplmentary to the main paper. \vspace is your friend.

- could you guys get any results on a long-video setup, it will greatly strengthen a paper, since action detectors like TubeR generally dont work beyond 64 frames to the best of my knowledge.
 - If long term video is not an option, then you could consider ucf, amir divided a long video into subclips i think when he made ucf101, and those are named as v_basketball_01, v_basketball_02 etc, you could in theory combine them , and show action detection on the long term video. mambas linear cost should allow you to fit those frames in memory, and tuber should fail. that will be a great selling point.

- please add related work w.r.t open world action detectors like SIA. (https://github.com/siatheindochinese/sia_act)

- how do you guys handle the cases where the action switches rapidly, like basketball dunk etc. i guess training a simple action classifier wont do the job.
- action detection is an inherently difficult problem, i.e. box around person has to be predicted only when the person is starting a action, how do you separate such cases from cases when the person is there in a frame and not doing an actual action (for eg, he is just dribbling basketball, and basketball dunk action has not yet started. i have seen a lot of action classifiers fail in such cases. an inductive bias to encode this constraint may be greatly helpful)

- **I could not find code in the supplementary**
- results should come before ablations.


[1] https://rccchoudhury.github.io/rlt/

--- my rating is 3, and i will change it once the experiments are more comprehensive. The official guidelines don't permit additional experiments, but without that the paper is incomplete. If those can be done, and added it would be good, otherwise, it might need another revision cycle. **It's a lot of work**, and i apologize for asking so many experiments, but that is a choice that authors might be willing to make. I will stand by my assessment.

---

> ### Author Rebuttal · Authors · 2025-07-30
>
> Really grateful! **They mean a lot to us.** Transformers've been everywhere, but we believe there’s room for new ideas, e.g. Mamba. Below are the point-by-point responses.
>
> **W1: how many encoder blocks…**
>
> R1: We use a pretrained 32-block Mamba encoder (32 justified by ablation in Supplementary A.5). However, the “glances only once” actually refers to our custom Decoder block (EQ-Mamba+QVI-Mamba+FFN), which is just ONE single layer. This setup lets us process all the encoded general spatio-temporal info in one single decoding pass. For decoder, our ablations show that just ONE layer already gets good results on datasets like JHMDB. We'll highlight clearly that it’s our newly designed, action-detection-specific Mamba-based Decoder that “glances only once".
>
> **W2: line 31: mamba is a good concept…**
>
> R2: **[New experiments added.]** We dug deep into resources around the IG dataset (moabitcoin/ig65m-pytorch). But it seems no public access to the *data*, only *pretrained weights* based on R(2+1)D_34.
>
> For TubeR using IG+K400, we find that TubeR sourced from the pretrained CSN-152 model. And CSN-152 comes from R(2+1)D_34 based on this paper by Facebook: *Video Classification with Channel-Separated Convolutional Networks*. Therefore, the key behind TubeR’s strong results is **not training on IG directly**, but starting from the pretrained WEIGHTS.
>
> That said, inspired by your suggestion, we tried other available large-scale pretraining sources beyond K400: Something-Something V2 (ssv2) and Breakfast-actions-dataset (breakfast), and integrated them into training.
>
> **ssv2**: We use 2 A40 GPUs and train for 50 epochs with the batch size of 30. And the fmap on the JHMDB is 64.1 (loss: epoch[0] 7.9834  -> epoch[49] 0.4847, grad_norm: epoch[0] 67.6360 -> epoch[49] 6.6854). **So K400's better than ssv2 in terms of pretrained model**.
>
> **breakfast:** 2 GPUs, 30 epochs with the batch size of 4. We tried to input 64 frames (initially, 8) for processing. And the fmap on the UCF101-24 is 42.0 (loss: epoch[0] 1.5368  -> epoch[29] 0.5196, grad_norm: epoch[0] 43.4297 -> epoch[29] 19.2594). map is not satisfactory, so we use K400 back but 64 frames. And the mAP improves to 63.43. **So pretrained model is vital and K400 is better than breakfast**.
>
> **W3: i think one baseline should be against flash attention…**
>
> R3: **[New experiments added.]** First, efficiency. To be completely transparent with you, our AI community, and us, we have to admit that the efficiency of our current mamba is not on par with the latest flash attention kernels. (Since our hardware doesn’t support flash attention v3, we use v2.8.2 released in July 2025.)
>
> We modified our decoder (as baselined in Table 11) and swapped in flash attention for Mamba. The decoder parameters dropped from 7.228M to 5.321M, and GFLOPs from 2.297 to 1.116.
>
>     class FlashDecoderLayer(...):
>         def __init__(self, ...):
>             super().__init__()
>             self.self_attn = MHA(...)
>             self.cross_attn = MultiheadAttention(...)
>             self.ffn = nn.Sequential(...)
>             self.norm = nn.LayerNorm(...)
>         def forward(self, tgt, memory):
>             tgt = self.norm(self.self_attn(tgt))
>             tgt, _ = self.cross_attn(tgt, memory, memory)
>             return self.norm(self.ffn(tgt))
>
> Sincerely hope our work can motivate more bottom-up optimization (e.g., by Tri Dao) to refine mamba C++/CUDA operators. Though still in its early stages, we're already able to show competitive efficiency compared to other transformer applications for this task.
>
> Second, accuracy. When we trained this flash attention for 30 epochs, the performance on JHMDB was poor, with fmAP50 at only 2.08(loss: epoch[0] 8.7587 -> epoch[29] 3.9476, grad_norm: epoch[0] 9.3831 -> epoch[29] 3.7795). This suggests maybe **simply** swapping in the implementation does not yield good results for our task.
>
> **W4: where are the results on ava...**
>
> R4: **[New experiments added.]** MOGO is initially designed for efficient, end-to-end detection. But AVA is for *multi-label for one bbox* tasks. Unlike standard AVA setups that require a separate bbox detector and multi-label classifier, MOGO predicts bbox and label(s) together in one stage. That’s why we only report mAP on the two datasets.
>
> But we attempted to **adapt** our decoder to enable *one bbox to multiple labels*, even without introducing an explicit cls branch. Also, loss function and other experiment settings are also changed accordingly. The simplified code is shown below:
>
>     class ...(...):
>         def forward(self, x):
>             mem, importance_logits = encoder(x)
>             hs = decoder(tgt, mem)
>             outs = [heads(output) for output in hs]
>             out = {
>                 'pred_actions': outs['class'], # whether a person/subject is detected (binary classification)
>                 'pred_boxes': outs['box'],
>                 'pred_logits': outs['obj'] # covering all classes in the dataset
>             }
>             return out
>
> In the original code, *class* (all classes+1) and *coords* were one-to-one matched. **On the AVA2.2, our modified MOGO achieved a val mAP of 16.2 (at step 26), which is comparable to CVRL-f32(SlowOnly-R50) with a mAP of 16.3. Notably, our decoder also demonstrates an advantage in GFLOPs (13.782 vs 42).**
>
> **W5: where are the vmap…**
>
> R5: **[New experiments added.]** In several SOTA works (e.g., Efficient Video Action Detection with Token Dropout and Context Refinement), **fmap50** is widely reported and we included it in Table 4. vmap evaluates the entire video as a tube and uses 3D IoU for evaluation, which is sensitive to tube generation quality. Nonetheless, we have included it.
>
> |Met|vmap20(U)|vmap20(J)|vmap50(U)|vmap50(J)|#Params|GFLOPs|Throughput|
> |-|:-:|:-:|:-:|:-:|:-:|:-:|:-:|
> |Ours|70.8|67.9|39.4|42.0|82M|102-104|256|
>
> Our vmap50 is relatively low compared to vmap20, mainly due to a few classes with low APs, which significantly affect the average: Basketball, BasketballDunk, TennisSwing, and VolleyballSpiking achieve nearly 0 AP at vmap50, despite reasonable scores at vmap20. This is because that our model is designed for action detection on the keyframe: take in a sequence, use the keyframe for spatial info, and our outputs are centered on those key moments, not continuous tubes.
>
> **W7: token selection blocks assumes a keyframe…**
>
> R7: **[New experiments added.]** We immediately **customized RLT-inspired token selection mechanism with our token importance scores**:
>
>     for patch_idx in range(tokens_per_frame):
>         prev = None
>         for i in range(num_frames):
>             idx = i * tokens_per_frame + patch_idx
>             t = memory[:, idx, :]
>             if prev is None or (t - prev).abs().mean() >= threshold:
>                 selected_memory.append(t)
>                 prev = t
>
> This block compresses repeated/unchanged tokens over time, and keeps semantically changed tokens. After 30 epochs of training, our model with RLT-based token selection achieved fmAP: 71.8 (Initial loss: 7.9931, grad_norm: 68.1378 (epoch 0), Final loss: 0.5432, grad_norm: 9.8033 (epoch 29)). Several hyperparameters (e.g., the threshold for rl, norm) remain tunable, and is promising after careful tuning.
>
> **W9&W10: could you guys get any results... & If long term video...**
>
> R11: **[New experiments added.]** We ran 3 new experiments on longer video sequences using UCF. For each setup, we used a matching pre-trained encoder and adjusted the decoder accordingly. Use 2×A40 GPUs with training epoch of 30.
>
> |Ex.|Frames|Pretrained Mode|Batch Size|mAP|
> |--|--|--|--|--|
> |1|16|K400|16|69.50|
> |2|64|Breakfast-actions-dataset|4|41.90|
> |3|64|K400|4|63.43|
>
> These results show that our method performs well on longer sequences. Due to limited time, we trained on only part of the data and for a relatively small epochs, so we expect mAP could improve further. We also observed that pretraining model is important, using K400 outperforms *breakfast* significantly. Also, our approach supports larger batch sizes than SOTA methods.
>
> **W12: how do you guys handle the cases…**
>
> R12: **[New exp added.]** We'll show that after processing with Mamba, **each token not only contains information about itself but also incorporates global information from all tokens**. So even the rapidly switched action info are also stored in each token in Mamba.
>
> (1) **Theory discussion**. In encoder:
>
> h(t) = A · h(t-1) + B · x(t)
>
> y(t) = C · h(t)
>
> The system state describes the **entire** system and will contain info from all tokens, meaning that each token will contain both its own information as well as global information from the whole system.
>
> More importantly, we implemented our encoder using bidirectional Mamba blocks. So it ensures that each token can access information from its **both sides** (preceding and following) from the entire system through the **bidirectional scan**.
>
> (2) **New experiment evidence**. We ran an experiment on JHMDB using only tokens with low importance scores. Even then, fmap50 only dropped from 76.7 to 72.7(during training, the loss decreased from 7.98 at epoch [0] to 0.55 at epoch [29], and the grad norm dropped from 67.6 to around 10). This shows after Mamba encoding, **even less “important” tokens contain enough global info for action detection**, though good token selection still helps accuracy.
>
> **W13: action detection is an inherently difficult…**
>
> R13: When training, we include a separate background class in addition to the dataset action categories. This is not only for handling empty/background objects, but also for *a person is present in the frame but not performing* if the data annotations are of high quality.
>
> **For W6 (add to limitation), W8 (format), W11 (add related work), W14 (code release), and W15 (content order), these are really helpful. and we’ll definitely address them in our final script! We hope you will kindly reconsider your evaluation in light of these updates.**

---

> ### Comment · Reviewer_Pwyx · 2025-08-01
> **some doubts (1/n)**
>
> dear authors :-), thanks for your rebuttal,
> can you tell some things:
> 1) **clarifications on novelty** re:r4, table 14 of videomamba paper, https://arxiv.org/pdf/2407.08476, shows results on action detection on ava 2.2, that is also a mamba backbone and obtains 22.1. the baseline mentioned in rebuttal (cvrl) gets 16.3, and mogo gets competitive 16.2. but, that still seems a large gap compared to 22.1, is there any particular reason that this comparison might be unfair? if its fair, there still seems a 5% significant gap that needs to be closed. video mamba is in **eccv 2024**, so its more than a year old already. i am sorry to have missed it earlier, since videomamba squeezed it in supplementary
>
> 2) re:r5,also, the rebuttal mentioned vmap 0.5 to be lower, since the method predicts on a keyframe, and. not optimized for continuity in the tubes. doesn't that appear to be a limitation? if its not continous that means its not very reliable?
>
> 3) re:r3 it is surprising to see flash attention giving mere 2.08. tuber results are better than mogo. what happens if tuber's attention is replaced by flash-attention. how does it compare against mogo on jhmdb?
>
> 4) re:r7, 71.8 is for which dataset? ucf or jhmdb? in any case, it is less than both the numbers in tab 4 of your paper. so is it not helping? does it perform worse than mogo's token selection mechanism? in your reply to reviewer **RYFL**,
>
> ``selected_memory.append(item['token']);
> memory = torch.stack(selected_memory, dim=0).transpose(0, 1)``
>
> you seem to be correctly using the selected tokens from the tokenizer, but not the run lengths. How are you processing the run lengths the tokenizer gives? Merely taking the tokens it selects won't work without using the run length also.
>
> 5) re:r13: action is a higher level concept. lets say a person is holding a basketball but not actually dribbling it. in other case, he is actually dribbling it. In both cases, the model will see both the person and the basketball. But, action happens in the latter not the former. Since perceptually, the inputs are the same, i.e. the person and the basketball, i dont believe that a mere classification head on top of bidirectional features will do the job: even if the annotations are of high quality.
>
> the problem boils down to how to model higher level actions? do you have any other insight into this :-)
>
> 6) lines 268 in the paper say: The encoder dominates the computational load counting for roughly 97% of the total FLOPs. It also seems to me that the encoder is processing all the tokens of all the frames. The token selection seems to be operating**after** the encoder, when mogo selects the top-k tokens, and only takes the encoder feature corresponding to those.
>
> however, methods like masked auto encoders, **first** mask image tokens, and only encode the tokens which are **not masked**. Is there any particular reason you didn't choose to do a similar thing, and instead **encode all the tokens prior to token-selection**?
>
> 7) **autoregressive frame decoders** you mentioned that mogo can process more frames than sota. there are methods like sam2, trackformer which autoregressively track/detect persons for more than 1000 frames. They dont do action detection, but action detection is just about detecting person and classifying actions, so in theory you could train them to do so. So any particular reason, that mogo is restricted to 64 frames. I guess the bottleneck comes from parallel processing of all frames, but SAM2 has no such limitation.

---

> ### Author Response · Authors · 2025-08-03
>
> Dear Pwyx, thanks for your comments. Please find our point-by-point responses below:
>
> **R1**: We revisited the paper and would like to clarify:
>
> (1) **No code or pretrained models are released** for the AVA results. Table 14 contains only a single-line result, with no detection-specific architecture or training details. Section 10.1 describes K400 recognition only, not detection.
>
> (2) **Table 14 is questionable**. The GFLOPs for their action detection model are exactly the same as those for their recognition model (34 GFLOPs, Table 14 vs. Tables 8 and 9). So whether action detection-specific modules (e.g., person localization, classification heads) were included in the FLOPs calculation? If not, the reported FLOPs may be significantly underestimated. We believe it is **not possible** to perform action detection simply using a video recognition model without any head.
>
> Our result is trained for 6 days (28 steps) on AVA v2.2. We acknowledge there is still room for further improvement.
>
> **R2**: Our method targets **keyframes** within a clip, metrics like fmap 0.5 may be a more appropriate evaluation.
>
> In our approach, when given an 8-frame clip, we select the **middle frame (e.g., frame 5)** as the keyframe for detection. The remaining non-keyframes provide important tokens (**Fig 4**) as **temporal context** and help model the action order across the clip. The model **performs detection only on the keyframe**.
>
> vmap is for generating continuous tubes across the entire clip(like tuber), **so it may underrepresent our strengths**. But we agree this is indeed a limitation.
>
> **R3**: [New experiment.] We carefully re-examined the code and made the correction: changed causal=True to causal=False in the MHA (self-attention in FlashDecoderLayer). In DETR-style object detection decoders, self-attention over object queries should be non-causal, as queries do not have a sequential order (unlike autoregressive).
>
> After this fix, we re-ran the experiments on 4 × A40 GPUs for 50 epochs with a batch size of 32. The map reached 70.1 (epoch[0] loss: 7.7654, grad norm: 14.1749 → epoch[49] loss: 0.4765, grad norm: 1.8073). Our approach indeed shows an advantage over the FlashAttention baseline in terms of map.
>
> **R4**: [New experiment.] Yes, in the previous version, we merely took the tokens. So this time we also embed the run-length of each selected token and add this to the corresponding token features before decoding to ensure that the temporal continuity info (how long a token remains stable) is explicitly incorporated in the model.
>
> **Also on JHMDB, using 4×A40 GPUs, training for 60 epochs with a batch size of 32. The final mAP reached 72.1 (epoch[0] loss: 8.4603, grad norm: 143.7739 → epoch[59] loss: 0.4385, grad norm: 2.4390)**. A small improvement over the previous 71.8, though not yet surpassing our best prior results.
>
> **R5**: Thank you. Our current approach does not employ an explicit “action switch” head as in TubeR, but instead uses a standard Hungarian matcher (DETR-style) with logits and bbox predictions. We also leverage temporal embeddings so the model can aggregate context across frames.
>
> Why does this DETR setup (with 2 heads) work here? Maybe can just view our method as an **image** action detection method. We target a keyframe (middle frame), while pulling in contextual tokens from the frames before and after. This helps the model perceive how a person’s action shifts across frames.
>
> Compared to tuber, we essentially replace its *action switch head* with our *background class*, which serves to make it efficient.
>
> However, this design may not explicitly capture precise action boundaries or subtle temporal transitions. For future work, we suggest: (1) integrating video localization task to help with start/end; (2) using point clouds and entropy for finer transition modeling.
>
> **R6**: Our pretrained encoder dominates the FLOPs, showing our **customized** decoder is lightweight.
>
> Why not mask/token selection before the encoder (like MAE)?
>
> (1) Our encoder is pretrained for a fixed number of tokens (frames × tokens/frame).
>
> (2) Reducing tokens before the encoder would break the expected input structure, may cause **unstable optimization and degraded performance**.
>
> Also, simply masking tokens without reducing their count does not reduce actual compute or inference time in most Transformer models.
>
> **R7**: Sorry for any confusion. Our claim that “mogo can process more frames” refers to GPU memory efficiency: in Fig. 7, MOGO uses less memory than SOTA, allowing larger batch sizes per GPU. Thus, MOGO can process more frames in practice.
>
> Regarding the 64-frame limit: our method (and TubeR, others) relies on pretrained models. The encoder we use (OpenGVLab/VideoMamba, HuggingFace) is trained for **at most** 64-frame inputs. mAP in action detection is sensitive to the quality and capacity of the pretrained encoder, so our limit comes from available pretrained models. But in theory, it can process long sequence.

---

> ### Comment · Reviewer_Pwyx · 2025-08-03
> **re:doubts (2/n)**
>
> dear authors,
>
> [1] thanks for looking into table 14, of videomamba paper.
> indeed, the flops reported seem identical to other tables,
> however,
> will you be able to add comparisons to ava in your final manuscript?
> also, videomae is still getting 22.5, which is 5% than mogos 16.2 , that is a remaining cause for concern.
>
> [2] other reviewers have also brought novelty concerns. my other comments are related to what could be done better to address those. can you confirm that these will be done, and if possible, do some additional analysis for those comments.
>
>
> ```The main concern of this paper is limited contribution to the community. Previous works have shown the effectiveness of Mamba on video tasks, this paper is more like a specific adaption of Mamba to video action detection.```
> I think you should highlight the computational advantages, and the importance of token importance block in your paper,
>
>
> ```The experiments are conducted on relatively short 8-frame clips, quite short video sequences. This experiment setting is not very convincing when considering Mamba's potential advantage on long sequences.```
>
> As you have mentioned, the best encoders are trained on $64$ frames, but i do think that a valid baseline of autoregressive decoders (like trackformer) would help establish good practices in this field. Would you be able to report at least one such baseline? I understand that it is not a standard experiment in the papers of spatio-temporal video action detection, but still asking.
>
> ```According to Table 4, the performance is still behind some Transformer-based models, which means a potential risk of trading peak performance for computation efficiency.```
>
> what happens if your token importance block is used in some transformer based method like TubeR.
>
>
> ```Narrow evaluation scope. Results are reported only on two small datasets (JHMDB 928 clips, UCF101-24 3k clips); the method is untested on large, benchmarks such as AVA or ActivityNet, where its linear-time claim would matter most.```
>
> You report results on AVA, which is appreciable.
>
> ```Token-importance mechanism insufficiently validated. Ablations vary k, but there is no comparison against random or heuristic token selection, so the incremental benefit of the learned importance loss remains speculative.```
>
> Thanks for performing the comparisons with rle baseline, its results seem lower than your token importance block, highlighting its efficiency.
>
> ```Depth-vs-performance anomaly. Table 2 shows a single-layer decoder achieves the best mAP, while a 6-layer stack collapses to ~32 mAP—counter-intuitive given the added capacity. Please clarify why deeper decoders “don’t work.” Is the drop caused by architectural issues (e.g., state-space over-smoothing or token interference) or by optimisation shortcomings (e.g., vanishing/exploding gradients, insufficient regularisation)? Diagnostics such as training-loss curves, gradient norms, or a variant that stabilises >1 layer would greatly illuminate whether this is a network-design or training limitation.```
>
> The overfitting logic seems sound.
>
> in summary:
> 1) can you add experiments on ava in main paper
> 2) add token importance block to some transformer based method, to show it improves performance rather than processing all tokens.
> 3) comment on how videomae is better by 5 points than mogo on ava, and what could be done to address those. a simple experiment .
> 4) highlight mogo is the first application of a linear time model in this area
> (5) maybe shift experiments before ablations.
>
> **This is my last set of doubts. I am satisfied with other responses. Please do clear them, and i will increase my score :-)**

---

> > ### Comment · Area_Chair_3fV1 · 2025-08-03
> >
> > Dear authors and reviewer Pwyx,
> >
> > It is great to see this discussion between you and clearing of some of the doubts. However, please note that while a paper can be revised for the final version, the expected changes should be limited and clear during the review process. "Major revisions" would be  tantamount to submitting a new paper which the reviewers would not have not had a chance to review.
> >
> > Your AC.

---

> > ### Author Response · Authors · 2025-08-06
> >
> > Dear Pwyx, thank you so much for your constructive responses:
> >
> > [1] Yes, we'd definitely like to add experiments on ava in main paper, including our training records.
> >
> > [2] Before presenting our new experiment progress, we want to highlight two **very very** important points:
> >
> > First, why we designed this token importance block for Mamba? Mamba lacks an attention map, unlike Transformers, which use attention scores to represent token importance. **Transformers motivated us to design token importance and we thought this is exclusively for Mamba (but we designed it lightweight and learnable)**. But when applying our token importance block **back** to Transformer-based methods, like TubeR, it somewhat introduces redundancy, as TubeR already has its own intrinsic attention-based token importance mechanism. Here, **we want to emphasize that our token importance system is primarily a tailored and innovative solution for Mamba**.
> >
> > Second, our token importance block is not an off-the-shelf, plug-and-play module, but a learning-based, multi-stage system **requiring end-to-end training**. The key steps are:
> >
> > (1) **Encoder-side importance calculation. The encoder includes a trainable MLP that outputs a continuous importance value for each token**, supervised by our custom loss (see (3)). In Fig 4 of our paper, we compare our improved, loss-supervised version with non-loss-supervised version.
> >
> > (2) **Token selection in the Decoder. The learned importance scores are used to select important tokens from the non-keyframes**.
> >
> > (3) **Loss design. The overall loss combines standard action detection loss** and token importance loss (bbox areas are considered important). The optimal loss ratio is determined through ablation studies.
> >
> > This system’s effectiveness relies on the synergy of all its components and joint training.
> >
> > When making it to other architectures like tubeR, key parameters, such as the loss ratio, may require extra tuning, and performance cannot be guaranteed without further adjustment, so if fully implemented, a new series of ablation is unavoidable. Therefore, we only implemented the **bolded part above** into tubeR. For these experiments, we ran two versions: one using the original TubeR implementation in our environment, and another with our token importance block added. **Both models were trained from scratch and now are ongoing**.
> >
> > **Also, we’d like to provide a theoretical evidence on using a token importance mechanism in Transformer-based methods can improve performance compared to processing all tokens.** In Efficient Video Action Detection with Token Dropout and Context Refinement (ICCV), see Figure 5, green line. However, their method relies solely on the transformer’s intrinsic attention scores. In contrast, our approach uses both the dynamic modeling of Mamba and a supervised loss. And we think ours *looks* a more effective token selection scheme, as shown in our Fig 4 vs theirs Fig 4.
> >
> > [3] We believe the main reason for the gap is the quality and maturity of available pretrained models. VideoMAE benefits from a large pool of well-optimized Transformer-based models with good masking strategies like high-ratio tube masking. We think that developing better Mamba-based **pretrained models** for video using **better masking strategies** will help close this gap. **Also, low-level CUDA/C++ optimization is important**.
> >
> > Regarding your suggestion of using an autoregressive decoder like TrackFormer, we explored this direction and agree it’s a promising future avenue. It actually aligns well with Mamba’s autoregressive nature, though there're still some implementation differences that would need to be addressed.
> >
> > [4] We’ll highlight that in main paper. The computational advantages (e.g., lower FLOPs, memory efficiency) and the token importance block will be highlighted.
> >
> > [5] We’ll shift the experiments section before the ablations as suggested, and also experiment results (1) ava; (2) random/heuristic token selection (rle); (3) longer sequence; (4) diagnostics such as training loss and gradient norm curves for layer > 1 are some good candidates.
> >
> > We thank you again for your valuable guidance!

---

> ### Comment · Reviewer_Pwyx · 2025-08-06
> **Reply and  change of rating**
>
> Dear Authors:
>
> Thank you so much for your detailed rebuttals, patience to my comments and additional experiments. I have increased my score accordingly. I would love to see your paper at NeurIPS, and am excited by how this might open up an area for linear-time spatio-temporal action detection.
>
> However, please note, as per **what the respected ACs mentioned**, i am not sure what constitutes as ```major revisions.```
>
> I would suggest categorizing your (promised) changes in a GLOBAL comment as:
> (1) Changes you have made which are just clarifications on the experiments **already** present in the paper.
> (2) Changes which are **brand new**, , and may be considered as ```substancial``` changes,  for eg, adding rle, ava experiments, discussions on table 14 of video-mamba paper not being reliable, flash attention baseline, discussions on training for >64 frames aka trackformer.
>
>
> To be fair in the review process, the ```ultimate decision lies with the respected AC, PC's who know far more```, and whether they will ```allow``` such substancial revisions,
>
> Thank you so much for your understanding,
>
> reviewer

---

> > ### Comment · Area_Chair_3fV1 · 2025-08-07
> >
> > Dear Authors,
> >
> > NeurIPS allows one extra page for accepted papers so this should allow you to include new experiments.
> >
> > Reviewer Pwyx's advice to list proposed changes in one place is a good one as it is hard to follow them through the discussion. It will allow all reviewers to make a judgment on whether the changes are acceptable and helpful.
> >
> > Please do not plan to include new results not presented in this forum as the reviewers have no way to evaluate them or their potential impact.
> >
> > AC

---

> ### Author Response · Authors · 2025-08-07
>
> Dear Pwyx, thank you very much for your comments! **For our list of proposed changes, please refer to our latest GLOBAL comment (Official Comment)**. We also welcome any further guidance or requirements from the AC and your side.
>
> Also, as we mentioned in [2] of *Replying to re:doubts (2/n)*, we have now completed the experiment of adding our token importance block into TubeR. Below is a summary of our findings on the JHMDB dataset, where we compared the standard TubeR (all tokens) with versions keeping 80% and 40% of non-keyframe tokens.
>
> [1] First, we observed that TubeR’s performance strongly depends on the provided pretrained weights (TubeR_CSN152_JHMDB.pth, maybe other can also work but we only tried this one). Without it, TubeR fails to train effectively on JHMDB (map at 0 after 20 epochs, despite the training loss decreasing). With the pretrained weights, TubeR reaches 71.2 map (consistent with TubeR’s GitHub Issues #14) after 1 epoch. Our focus is on the first epoch because we already used the final weights, TubeR_CSN152_JHMDB.pth.
>
> [2] With token reduction based on our importance block, the results are as follows:
>
> - 80% non-keyframe tokens: mAP = 72.05 (loss: 9.279)
>
> - 40% non-keyframe tokens: mAP = 72.0 (loss: 10.02)
>
> These results are comparable to the full-token baseline, showing that our token importance block maintains accuracy even when fewer tokens are used. **Our small-scale experiments indicate that our token importance block is not only effective for our keyframe detection, but also does not lead to a map drop for multi-frame detection tasks like TubeR**, which is encouraging.
>
> Thank you again for your valuable suggestions. Please let us know if you require further clarification.
>
> Best regards

---

> ### Author Response · Authors · 2025-08-07
>
> Dear AC,
>
> Thank you very much for your very helpful reminders.
>
> We have already presented our list of proposed changes **in the GLOBAL comment as you kindly suggested**. We'll continue to follow your advice and ensure all changes are compliant.
>
> Thank you again for your support.
>
> Best regards

---

> > ### Comment · Reviewer_Pwyx · 2025-08-07
> > **Wrapping up and End of Discussions :-)**
> >
> > Respected Authors,
> >
> > Thanks for your comments, these changes look fine to me.
> >
> > As we near the end of the discussions,
> >
> > As they say in vulcan,
> >
> > 🖖
> >
> > ```Live long and Prosper```
> >
> > reviewer

---

> > > ### Author Response · Authors · 2025-08-08
> > >
> > > Dear Reviewer Pwyx,
> > >
> > > Many many thanks for your feedback and support, and for the Vulcan wisdom! 🖖
> > >
> > > Wishing you a prosperous and warp-speed research journey as well!!
> > >
> > > Best regards

---

### Official Review · Reviewer_RyfL · 2025-07-05

**Clarity:** 3
**Significance:** 3
**Originality:** 3
**Rating:** 4
**Confidence:** 4

**Summary:**

The paper proposes MOGO, a fully Mamba-based  framework for spatio-temporal action detection, which consists of a mamba encoder models cross-frame context, and two novel state-space modules: EQ-Mamba (query self-refinement) and QVI-Mamba (query–video fusion). Evaluations on JHMDB and UCF101-24 benchmarks indicate the effectiveness of the proposed models.

**Questions:**

- **Questions for Figure-5.** In Fig. 5, mAP rises up to k ≈ 40% and then drops sharply as k increases further, even though the decoder has more visual information. Why does retaining a larger fraction of tokens hurt performance? Please give a principled explanation and, if possible, add an ablation that separates key-frame vs. non-key-frame tokens to pinpoint the root cause.

- **Extreme sensitivity to the triplet weight w (cls : box : ovl)**.
In Table 3(a) the mAP plummets from 68 → 4 (!) when switching from 5 : 1 : 1 to 1 : 5 : 2, even though the overall loss is still a weighted sum of the same three terms. I wonder what failure mode causes such a catastrophic drop?

- **Non-monotonic behaviour of the total-loss ratio λ.**
Table 3(b) shows mAP rising from 73.3 → 75.6 at λ = 0.5, then falling to 74.4 for λ ≥ 1. Why does a smaller λ (i.e., weaker auxiliary loss) help, but an even smaller 0.1 hurts? Is there an intuition—e.g., regularisation vs. under-training—that could guide practitioners in picking λ without an exhaustive sweep?

**Ethical Concerns:**

["NO or VERY MINOR ethics concerns only"]

**Final Justification:**

Thanks for the response. The rebuttal is thorough and adds convincing evidence.

- Novelty: Still moderate, but clarified as the first purely Mamba-based detector with a new token-importance loss.

- Evaluation: New AVA2.2 results strengthen the scope and highlight efficiency.

- Token importance: Random/heuristic ablations confirm the benefit of the proposed mechanism.

- Depth issue & loss weights: Additional training curves and logs provide plausible explanations.

Overall, concerns are largely addressed. I raise my score from 3 → 4 (Borderline accept) since the empirical validation is now much stronger and efficiency advantages are clear.

**Limitations:**

yes

**Quality:**

3

**Strengths And Weaknesses:**

**Paper Strengths**

- The paper is overall well-structured and easy to follow. The authors provide clear motivation, detailed methodology, and thorough experimental validation.
- Large efficiency gains: 2–3× lower GFLOPs, latency (256 img/s) and GPU memory than strong Transformer baselines while keeping competitive mAP
- The authors provide well-documented ablations (decoder depth, token-retention ratio, loss weights.

**Weaknesses**
- Limited novelty. Prior work has already demonstrated Mamba or other state-space models on video classification and temporal modeling, so substituting Mamba for the Transformer decoder feels incremental; the contribution lies more in engineering than in fundamentally new theory or tasks.
- Narrow evaluation scope. Results are reported only on two small datasets (JHMDB 928 clips, UCF101-24 3k clips); the method is untested on large,  benchmarks such as AVA or ActivityNet, where its linear-time claim would matter most.
- Token-importance mechanism insufficiently validated. Ablations vary k, but there is no comparison against random or heuristic token selection, so the incremental benefit of the learned importance loss remains speculative.
- Depth-vs-performance anomaly. Table 2 shows a single-layer decoder achieves the best mAP, while a 6-layer stack collapses to ~32 mAP—counter-intuitive given the added capacity. Please clarify why deeper decoders “don’t work.” Is the drop caused by architectural issues (e.g., state-space over-smoothing or token interference) or by optimisation shortcomings (e.g., vanishing/exploding gradients, insufficient regularisation)? Diagnostics such as training-loss curves, gradient norms, or a variant that stabilises >1 layer would greatly illuminate whether this is a network-design or training limitation.

---

> ### Author Rebuttal · Authors · 2025-07-29
>
> We are grateful for your comments. Below, we address each of the points you identified.
>
> **[Weaknesses addressed]**
>
> **W1: Limited novelty. Prior work...**
>
> R1: Thank you for your feedback. While we agree that prior work has applied Mamba and other state-space models to video tasks, we respectfully clarify the following key points:
>
> (1) To the best of our knowledge, our work is **the first to present a pure Mamba-based architecture** for efficient video action detection (see Sec. 2.2).
>
> (2) **The token construction mechanism with learnable importance** enables effective selection of spatio-temporally relevant tokens (Sec. 2.3). Also, our newly designed loss function effectively distinguishes between important and unimportant tokens (see **Figure 4**).
>
> (3) Our approach achieves strong results in terms of **params, FLOPs, memory, and inference speed**, outperforming or matching Transformer-based SOTA methods (see Table 4, Figure 7).
>
> **W2: Narrow evaluation scope. Results are reported...**
>
> R2: **[New experiments added.]** Our proposed MOGO was originally designed as a lightweight model, focusing on efficient action detection, rather than the *one bbox to multiple label* tasks such as AVA. In standard AVA evaluation, the typical approach involves building a bbox detector (nearly all are based on R-CNN), followed by a separate multi-label classification branch for each detected bbox. However, our MOGO is inherently end-to-end, and each bbox is directly associated with its predicted label(s) in a single stage, without the need for an additional block (No R-CNN and classification blocks). This is why we only report mAP on the two datasets.
>
> Nevertheless, we attempted to **adapt** our decoder to enable *one bbox to multiple labels* without introducing an extra classification branch. Also, loss function and other experiment settings are also changed accordingly. The simplified code is shown below:
>
>     class ...(...):
>         def forward(self, x):
>             mem, importance_logits = encoder(x)
>             hs = sdecoder(tgt, mem)
>             outs = [self.heads(output) for output in hs]
>             out = {
>                 'pred_actions': outs['class'], # whether a person/subject is detected (binary classification)
>                 'pred_boxes': outs['box'],
>                 'pred_logits': outs['obj'] # covering all classes in the dataset
>             }
>             return out
>
> In the original code, class (all classes+1) and coords were one-to-one matched. **On the AVA2.2, our modified MOGO achieved a val mAP of 16.2 (at step 26), which is comparable to CVRL-f32(SlowOnly-R50) with a mAP of 16.3. Notably, our decoder also demonstrates an advantage in GFLOPs (13.782 vs 42).**
>
> **W3: Token-importance mechanism insufficiently validated. Ablations vary k...**
>
> R3: **[New experiments added.]** Thank you for raising this insightful question. We answer your question by 1) previous research evidence and 2) new experiments.
>
> First, according to a prior work, *Efficient video action detection with token dropout and context refinement* (ICCV), **selecting non-background tokens as important tokens leads to good performance on action detection tasks**. This provides theoretical support for our token-importance mechanism.
>
> Second, according to your recommendation, we included new experiments for comparison.
>
> **(1) Random token selection**. We revised our code as following:
>
>     for each frame in video:
>         frame_tokens = get_tokens(memory, frame)
>         selected_indices = random_select(top_k = 40% of frame_tokens)
>         selected_tokens = frame_tokens[selected_indices]
>         add selected_tokens to selected_memory
>     final_memory = concat(selected_memory)
>
> We removed the *important logits* part and randomly selected tokens for this experiment. After 30 epochs of training, our model achieved fmAP: **67.2** (Initial loss: 7.9832 (epoch 0), Final loss: 0.6982 (epoch 29)). Compare to the benckmark (mAP: **67.4**) with the same environment and setting in Figure 5, our initial method shows superior to the random method.
>
> **(2) Heuristic token selection**. We refer to *Faster Video Transformers with Run-Length Tokenization* and integrated this heuristic RLT-inspired token selection mechanism into our pipeline, and customized it with our token importance scores:
>
>     for patch_idx in range(tokens_per_frame):
>         ...
>         for i in range(frame_number):
>             start_idx = i * tokens_per_frame
>             token = memory[:, start_idx + patch_idx, :]  # [B, D]
>             # compare to previous frame's token
>             if prev_token is not None:
>                 diff = (token - prev_token).abs().mean(dim=-1)  # [B]
>                 if (diff < threshold).all():
>                     run_length += 1
>                     continue
>             if run_length > 0:
>                 patch_tokens[-1]['run_length'] = run_length + 1
>             patch_tokens.append({'token': token, 'run_length': 1})
>             ...
>         if run_length > 0:
>             patch_tokens[-1]['run_length'] = run_length + 1
>         for item in patch_tokens:
>             selected_memory.append(item['token'])
>             run_length_embeds.append(item['run_length'])
>     memory = torch.stack(selected_memory, dim=0).transpose(0, 1)
>
> This block compresses repeated/unchanged tokens over time, and only keeps tokens when *semantic changes* are detected. After 30 epochs of training, our model with RLT-based token selection achieved fmAP: **71.8** (Initial loss: 7.9931, grad_norm: 68.1378 (epoch 0), Final loss: 0.5432, grad_norm: 9.8033 (epoch 29)). **Compare to the benckmark (mAP:76.7) with the same environment and setting, our method shows superior to this heuristic method and can show the benefit of our learned importance loss we proposed.**
>
> **W4: Depth-vs-performance anomaly. Table 2 shows a single-layer...**
>
> R4: Due to the rebuttal policy, we cannot display figures, but we summarized the training loss, gradient norms, and mAP across all >1-layer decoder experiments in the following. Those experiments were all conducted for 50 epochs with a batch size of 32 and 1 A40 GPU. For 3 decoder layers, the training loss 5.1019 -> 0.4732, grad norm: 87.6617-> 5.8463,with mAP reaching 0.710. However, for 6 layers, loss 4.9990 -> 2.6747 **decreases but plateaus at a higher value**, grad norm: 49.6281 -> 5.2908 with mAP saturating 0.32.
>
> Our explanation is that when the decoder has fewer layers, the network learns stably, loss converges, and the model achieves good performance. This suggests that the network capacity is sufficient to model the task. However, when the decoder is deeper (e.g., 6 layers), we observe that while the training loss still decreases, it plateaus at a much higher value. This is likely to indicate the dataset size is limited and the model is overfitting with more layers.
>
> **[Questions addressed]**
>
> **Q1: Questions for Figure-5. In Fig. 5...**
>
> R5: **[New experiments added.]** Thank you for bringing up this question. We still answer this question by 1) the evidence from existing works, and 2)  new experimental evidence.
>
> First, **similar trends have been observed in prior research**. In Figure 5 (the green line) of Efficient Video Action Detection with Token Dropout and Context Refinement (ICCV), we can see keeping more tokens doesn’t always mean better results.
>
> Second, **k is all about non-keyframe tokens**. Our method is to keep **all the key frame tokens** and **the most important tokens from those non-keyframes** (like the main subject or person) but **only detect the key frame**. The non-keyframes are mainly to help provide some extra context. Therefore, according to your suggestion, we aim to show that **including 100% of non-keyframe tokens may dilute the representation of keyframe tokens**.
>
> | Case                                              | mAP  |
> |:-|:-|
> | Only non-keyframe                                      | 53.7 |
> | Proposed (key-frame + non-keyframe)        | 69.0 |
>
> We conducted a set of mini-experiments using a small batch size and similar settings, comparing the use of all non-keyframe tokens with our proposed method. The results show that the approach using 100% of non-keyframe tokens yields lower mAP compared to our proposed method.
>
> In addition, when we split a video into tokens, not every token is **equally helpful** for action detection. For example, many background tokens may add noise and distract the model. We further conducted an experiment that only used low importance tokens, **the mAP drops noticeably (76.7->72.7)** (during training, the loss decreased from 7.98 at epoch [0] to 0.55 at epoch [29], and the gradient norm dropped from 67.6 to around 10).
>
> **Q2: Extreme sensitivity to the triplet weight...**
>
> R6: **[New experiments added.]** To answer your question, we show the experiment records of both the original 1:5:2 run and 5:1:1 baseline for reference. And **we conducted the 1:5:2 experiment once again (a repeat 1:5:2 (20250728))** and found a similar trend.
>
> When using 1:5:2 weights, we observed high loss (epoch[0] 6.0x/5.9x -> epoch[29] 2.x) and grad norms > 400 after 20 epochs, while mAP stayed under 0.05. In contrast, with 5:1:1, loss:epoch[0]: 8.0x -> epoch[29] 0.8x, grad norms drops steadily after 10 epochs, and mAP climbs above 0.6. So now we have enough evidence to say that the 1:5:2 setting may overwhelm optimization with box/ovl weights, preventing the model from learning effective action features.
>
> **W3: Non-monotonic behaviour of the total-loss ratio λ...**
>
> R7: Our loss is custom-designed: it combines a **standard action detection loss** and our **own token importance loss**. Since these two terms **may differ in scale**, tuning λ is necessary. Also, mAP differences across λ values falls within an acceptable range, which is common for such hyperparameters. We found that λ = 0.5 gives the best trade-off on this dataset.
>
> **We hope you will kindly reconsider your evaluation in light of these updates.**

---

> ### Author Response · Authors · 2025-08-06
>
> Dear RyfL,
>
> Thank you again for your constructive review! We wanted to follow up and let you know that we've carefully addressed the main concerns you raised, and have **conducted several new experiments, such as token-importance mechanism and experiments on new dataset**, based on your suggestions.
>
> If you have any additional questions, please let us know, and we're more than happy to discuss further!
>
> Thank you very much for your time and attention!

---

> ### Author Response · Authors · 2025-08-08
>
> Dear Reviewer RyfL,
>
> Thank you again for your earlier feedback! It was very helpful in guiding our revisions. **We would like to kindly inform you that we have now updated all your concerns in our latest GLOBAL official comment**, including the following updates:
>
> ---
>
> **Regarding Weakness [1]**:
>
> - **Line ~75, main**: Highlight the computational advantages and the importance of the token importance block. Also, we will emphasize that our method is the first to present a pure Mamba-based architecture, achieving strong results in terms of parameters, FLOPs, memory, and inference speed, while outperforming or matching Transformer-based SOTA methods.
>
> **Regarding Weakness [2]**:
>
> - **Line 0, extra**: Add AVA results, implementation details, and discussions.
>
> **Regarding Weakness [3]**:
>
> - **Line 0, extra**: Add heuristic token selection (RLE) and random token selection for comparison.
>
> **Regarding Weakness [4]**:
>
> - **Line 0, extra**: Add diagnostics, including training loss and gradient norm curves for layers >1.
>
> ---
>
> **Regarding Question [1]**:
>
> - **Line ~79, main**: Clarify our task more clearly: keyframe action detection in clips (detects the keyframe and uses non-key frames for reference). We’d like to emphasize this. In our task, we detect only the one keyframe given multiple frames (8 frames in total, but only detect the middle one frame). So not all tokens from other frames (non-keyframe) are useful. They may represent background info which may serve as redundancy. However, retaining object tokens (such as human) from those other frames is crucial, as these provide valuable cues for the model to recognize the sequence and order of actions.
>
> - **Line ~123, main**: Theoretical evidence (Efficient Video Action Detection with Token Dropout and Context Refinement (ICCV), Figure 5) on using a token importance mechanism can improve performance compared to processing all tokens.
>
> - **Line 0, extra**: Add experiments integrating our token importance block into TubeR, showing improvements in small-scale tests.
>
> **Regarding Questions [2] & [3]**:
>
> - **Line 0, extra**: Add diagnostics.
>
> - **Line ~247, main**: Clarify that due to our redesign of the loss function, the relationships between components have changed, making results sensitive to weight choices; re-evaluation of weight configurations is therefore necessary for optimal performance.
>
> ---
>
> Should there be anything else you’d like to discuss or clarify, we would be more than happy to respond promptly.
>
> Best regards

---

### Official Review · Reviewer_Ambp · 2025-07-06

**Clarity:** 3
**Significance:** 2
**Originality:** 2
**Rating:** 4
**Confidence:** 3

**Summary:**

This paper proposes a Mamba-based framework for video action detection. It includes a pure Mamba-based encoder-decoder with better computational efficiency than Transformers and a video token construction mechanism that selects the most relevant spatio-temporal tokens. The proposed method demonstrates good performance in terms of efficiency and action detection ability on the JHMDB and UCF101-24 datasets.

**Questions:**

I have two questions about the robustness of token construction mechanism:
- It applies a fixed top k percentage of tokens. How does this fixed-rate strategy handle videos with high action density that require more and more tokens?
- According to Table 2(c) and Figure 5, when k is above 40, the performance will decrease. This is very counter-intuitive. Do the authors have some insight about it ?

**Ethical Concerns:**

["NO or VERY MINOR ethics concerns only"]

**Final Justification:**

Thank authors for the detailed rebuttal. Besides the trade-off between peak performance and efficiency, most of my concerns have been addressed. I will increase my score to weak accept.

**Limitations:**

Yes, the limitations are discussed in Section 4, Conclusion and Future Work.

**Quality:**

3

**Strengths And Weaknesses:**

Strengths
- The introduction of EQ-Mamba and QVI-Mamba enables Mamba's capabilities to the task of video action detection.
- The proposed method demonstrates superior performance in terms of GFLOPs, inference speed, and memory usage compared to Transformer-based methods.

Weaknesses
- The main concern of this paper is limited contribution to the community. Previous works have shown the effectiveness of Mamba on video tasks, this paper is more like a specific adaption of Mamba to video action detection.
- The experiments are conducted on relatively short 8-frame clips, quite short video sequences. This experiment setting is not very convincing when considering Mamba's potential advantage on long sequences.
- According to Table 4, the performance is still behind some Transformer-based models, which means a potential risk of trading peak performance for computation efficiency.

---

> ### Author Rebuttal · Authors · 2025-07-29
>
> Thank you so much for your review. Below, we address each of the Weaknesses and Questions you mentioned.
>
> **[Weaknesses addressed]**
>
> **W1: The main concern of this paper is limited contribution to the community. Previous works have shown the effectiveness of Mamba on video tasks, this paper is more like a specific adaption of Mamba to video action detection.**
>
> R1: MOGO is re-designed from the ground up for Mamba. We provide both motivation (Fig. 1, Sec. 2.2) and step-by-step method evolution (Sec. 2.4), showing how we restructured each part of the detection pipeline based on the capabilities of Mamba (state-space modeling, linear-time sequence processing), rather than directly borrowing prior approaches. We respectfully clarify the following key points:
>
> (1) To the best of our knowledge, our work is the first to present an end-to-end, **pure Mamba-based architecture** specifically tailored for efficient video action detection, without reliance on Transformer or RCNN-like modules (see Sec. 2.2).
>
> (2) **The video token construction mechanism with learnable token importance** enables effective selection of spatio-temporally relevant tokens (Sec. 2.3). This is not present in previous Mamba-based video works.
>
> (3) Our newly designed loss function effectively distinguishes between important and unimportant tokens (**see Figure 4**). This allows the decoder to receive more informative inputs while also compressing the video tokens.
>
> (4) Our approach achieves strong results in terms of **params, FLOPs, memory, and inference speed**, outperforming or matching Transformer-based SOTA methods with significantly lower resource usage (see Table 4, Figure 7).
>
> **W2: The experiments are conducted on relatively short 8-frame clips, quite short video sequences. This experiment setting is not very convincing when considering Mamba's potential advantage on long sequences.**
>
> R2:  **[New experiments added.]** Thank you and indeed, previous experiments were based on 8-frame clips. So we conducted 3 new experiments on longer video sequences and report the results on UCF101-24. For each setup, we used the corresponding pre-trained encoder (matching the number of frames), and adjusted the decoder settings accordingly. The results are summarized below:
>
> |Ex.|#GPUs|Frames|Pretrained Model|Batch Size|Epoch|mAP|
> |--|--|--|--|--|--|--|
> |1|2×A40|16|K400|16|30|69.50|
> |2|2×A40|64|Breakfast-actions-dataset|4|30|41.90|
> |3|2×A40|64|K400|4|30|63.43|
>
> These results demonstrate that our method performs well on longer sequences. **Due to limited time, we trained on only part of the data and for a relatively small epochs, so we expect mAP could improve further with more training.** We also observed that the choice of pretraining model is important, using K400 outperforms breakfast significantly for the same 64-frame setup. One advantage of our approach is that it supports larger batch sizes than SOTA methods, which is beneficial for efficiency. Other training metrics are shown for your reference:
>
> 1. loss: epoch[0] 1.4113  -> epoch[29] 0.3047, grad_norm: epoch[0] 81.2901 -> epoch[29] 4.8639
>
> 2. loss: epoch[0] 1.5368 -> epoch[29] 0.5196, grad_norm: epoch[0] 43.4297 -> epoch[29] 19.2594
>
> 3. loss: epoch[0] 1.5457 -> epoch[29] 0.3382 , grad_norm: epoch[0] 38.0396 -> epoch[29] 9.7863
>
> **W3: According to Table 4, the performance is still behind some Transformer-based models, which means a potential risk of trading peak performance for computation efficiency.**
>
> R3: Yes, we acknowledge this observation. However, although our current Mamba implementation is still at an early stage, we are already able to demonstrate **the best efficiency performance** and competitive precision compared to Transformers, and clear advantages over traditional CNN-based approaches on this downstream task.
>
> The strong performance of Transformer-based models relies on two important factors: (1) the availability of a very large pool of pretrained models, and (2) highly optimized low-level C++/CUDA implementations.
>
> Currently, due to the limited research on video using Mamba, it does not yet benefit from such extensive pretraining resources or mature hardware/software optimizations, which inevitably impacts peak performance. That said, we think that **Transformers May Not Be The Only Possible Foundation For Future AI**. We sincerely hope our work can motivate more researchers to explore novel low-level architectures and drive further improvements in the AI community.
>
> **[Questions addressed]**
>
> **Q1: I have two questions about the robustness of token construction mechanism: It applies a fixed top k percentage of tokens. How does this fixed-rate strategy handle videos with high action density that require more and more tokens?**
>
> R4: **[New experiments added.]** Thank you for raising this insightful question. We answer this question by 1) theory discussion of our Mamba implementation, and 2) new experimental evidence, in order to show that after processing with Mamba, **each token not only contains information about itself but also incorporates global information from all tokens**. Therefore, the action density in the video and the token keep rate have no longer to be strictly correlated.
>
> First, those fixed top-k percentage of tokens are used as decoder input, and the decoder receives tokens from the encoder. In the encoder, we employ a Mamba implementation:
>
> h(t) = A · h(t-1) + B · x(t)
>
> y(t) = C · h(t)
>
> It tells that the system state describes the **entire** system, so it contains information from all tokens. This means that each individual token (in our system with shape [N, D]) will contain both its own information as well as global information from the whole system. More importantly, we implemented our encoder using **bidirectional Mamba blocks**. Therefore, this design ensures that each token can access information **from its both sides (preceding and following)** from the entire system **through the bidirectional scan**.
>
> Second, in a new experiment on the JHMDB dataset with a similar setup, we included only tokens with low importance scores. We observed that fmap50 dropped from 76.7 to 72.7 (during training, the loss decreased from 7.98 at epoch [0] to 0.55 at epoch [29], and the gradient norm dropped from 67.6 to around 10). Below is the code:
>
>     for i in range(frame_number):
>         start_idx = i * tokens_per_frame
>         end_idx = (i + 1) * tokens_per_frame
>         frame_memory = memory[:, start_idx:end_idx, :]
>         frame_importance_logits = importance_logits[:, start_idx:end_idx]
>         top_k = int(tokens_per_frame * 0.4)
>         _, indices = torch.topk(frame_importance_logits, top_k, dim=1, largest=False, sorted=False)
>         selected_frame_memory = torch.gather(frame_memory, 1, indices.unsqueeze(-1).expand(-1, -1, D))
>
>         selected_memory.append(selected_frame_memory)
>     memory = torch.cat(selected_memory, dim=1)
>
> These results show that even background tokens, those originally considered unimportant, can still yield reasonable action detection performance after being processed by the encoder. **This suggests that each token indeed contains global information.** However, the noticeable drop in mAP also demonstrates the importance of proper token selection.
>
> In summary, Mamba’s intrinsic scan mechanism allows each token to aggregate information across the sequence, including high-density actions. This is a key advantage of our approach.
>
> **Q2: According to Table 2(c) and Figure 5, when k is above 40, the performance will decrease. This is very counter-intuitive. Do the authors have some insight about it?**
>
> R5: **[New experiments added.]** Thank you for bringing up this question. We still answer this question by 1) the evidence from existing works, and 2)  new experimental evidence.
>
> First, **similar trends have been observed in prior research**. In Figure 5 (the green line) of Efficient Video Action Detection with Token Dropout and Context Refinement (ICCV), a similar non-monotonic pattern shows up: keeping more tokens doesn’t always mean better results.
>
> Second, **k is all about non-keyframe tokens**. Our method is to keep **all the key frame tokens** and **the most important tokens from those non-keyframes** (like the main subject or person) but **only detect the key frame**. The non-keyframes are mainly to help provide some extra context, helping the model figure out the order or direction of the action. Therefore, in the following new experiment, we aim to demonstrate that **including 100% of non-keyframe tokens may dilute the representation of keyframe tokens**.
>
> | Case                                              | mAP  |
> |:--------------------------------------------------|:-----|
> | Only non-keyframe                                      | 53.7 |
> | Proposed (key-frame + non-keyframe)        | 69.0 |
>
> We conducted a set of mini-experiments using a small batch size and similar settings, comparing the use of all non-keyframe tokens with our proposed method. The results show that the approach using 100% of non-keyframe tokens yields lower mAP compared to our proposed method.
>
> In addition, when we split a video into tokens, not every token is **equally helpful** for action detection. For example, many background tokens, even though including global info, may add noise and distract the model. As shown in Q1's experiment, if we only use low importance tokens (mostly background tokens), **the mAP will drop noticeably (76.7->72.7)**.
>
> **Thank you so much and we hope you will kindly reconsider your evaluation in light of these updates.**

---

> ### Author Response · Authors · 2025-08-06
>
> Dear Ambp,
>
> Thank you again for your thoughtful review! We wanted to follow up and let you know that we've carefully addressed the main concerns you raised, and have **conducted several new experiments, such as longer video sequences and the robustness of token construction mechanism**, based on your suggestions.
>
> If you have any additional questions, please let us know, and we're more than happy to discuss further!
>
> Thank you very much for your time and attention!

---

> ### Comment · Reviewer_Ambp · 2025-08-06
>
> Thank authors for the detailed rebuttal. Most of my concerns have been addressed. Although the proposed method is still behind some Transformer-based models and the trade-off between peak performance and efficiency is still an issue, I will increase my score.

---

> > ### Author Response · Authors · 2025-08-07
> >
> > Dear Ambp,
> >
> > Thank you very much! We really appreciate your valuable feedback and we will continue to work on this in our future research.
> >
> > Thank you again for your time and valuable comments.
> >
> > Best regards

---

### Comment · Reviewer_Pwyx · 2025-08-06

Dear Authors,
If you have any other experiments for me, please do let me know,
However, based on the discussions with AC, i am not yet sure, how much of the changes are accepted as "Major Revisions"
I will be grateful for the chance to discuss more in the AC-Reviewer time period,

Based upon that, i ill decide adjust my rating accordingly,
Thanks so much for your time,

best,
reviewer

---

> ### Author Response · Authors · 2025-08-06
>
> Dear Pwyx,
>
> Thank you very much for your attention! Following your suggestions in the last discussion cycle, we are currently running experiments on integrating the token importance block into tuber. Aside from this ongoing experiment, we think we've addressed all other points you raised, **as shown in our latest response**.
>
> Truly appreciate all your constructive feedback and we've **done our best to strengthen the paper** based on your suggestions. We **sincerely** hope our revisions will meet your expectations! Thanks again!

---

### Author Response · Authors · 2025-08-07

Dear AC and Reviewers,

Thank you very much for your valuable feedback, constructive suggestions, and support throughout the review process. Although conducting additional experiments has been challenging, **your input has truly motivated us and helped strengthen our paper.** As suggested, we summarize all our proposed changes (**as candidates, and we welcome any further guidance or requirements from the AC and Reviewers**):

**Notation:**
- *main*: main paper
- *supp.*: supplementary material
- *extra*: additional 1-page extension allowed by NeurIPS for accepted papers (as kindly reminded by the AC)
- *~*: approximately (e.g., Line ~71 = around line 71; final version may shift slightly)

**[1] Changes we have made which are just clarifications on the experiments already present in the paper**
- **Line ~20, main:** Add code release URL.
- **Line ~75, main:** Highlight the computational advantages and the importance of the token importance block.
- **Line ~79, main:** Clarify our task more clearly: *keyframe* action detection in clips (detects the keyframe and uses non-key frames for reference). **We’d like to emphasize this. In our task, we detect only the one keyframe given multiple frames (8 frames in total, but only detect the middle one frame). So not all tokens from other frames (non-keyframe) are useful. They may represent background info which may serve as redundancy. However, retaining object tokens (such as human) from those other frames is crucial, as these provide valuable cues for the model to recognize the sequence and order of actions. Therefore, a well-designed selection of important tokens is essential.** On the other hand, for tasks that require detection of all frames rather than just one keyframe, like TubeR, it is not guaranteed that reducing the number of tokens will always improve map, as every frame is to be detected, so all frames are keyframes.
- **Line ~123, main:** Theoretical evidence (Efficient Video Action Detection with Token Dropout and Context Refinement (ICCV), Figure 5) on using a token importance mechanism can improve performance compared to processing all tokens.
- **Line ~166, main:** Clarify the WHOLE system of the token importance block. (a multi-stage system requiring end-to-end training)
- **Line ~247, main:** Clarify that due to our REDESIGN of the loss function, the relationships between the different components have also changed, making the final results sensitive to the choice of weights. Therefore, a re-evaluation of the weight configurations is necessary to optimize performance.
- **Line ~307, main:** Explain why TubeR using its pretrained model perform well. TubeR uses CSN-152 weights, originally from R(2+1)D_34 using IG. The key is access to strong pretrained weights. Mamba does not have such a pretrained pool, which may explain the difference. Thus, developing pretrained models for Mamba is important.
- **Line ~328, main:** Add a limitation: fixed query, our method cannot handle >100 people.
- **Line ~438, supp.:** Add explanation for “why only glances once” (our action-detection-specific Mamba-based decoder glances only once).
- **Line ~440, supp.:** Add related work: siatheindochinese/sia_act.
- **Line ~515, supp.:** Clarification on v-map50 (since ours is the **keyframe** action detection for efficiency).
- **Line 0, extra:** Add diagnostics: **training loss and gradient norm** for layers >1.
- **Line 0, extra:** Discuss training with >64 frames, including TrackFormer details (autoregressive training, masking).
- **Formatting:** shifted figures, vspace. Reordered content: main results before ablation studies.

---

**[2] Changes which are brand new**

- **Line ~545, supp.:** Add results with other pretraining sources (SSV2, Breakfast-actions-dataset) beyond K400.
- **Line 0, extra:** **Heuristic token selection**: rle (version 2, considering run length), **Random token selection**.
- **Line 0, extra:** Add AVA results, implementation details, and discussions (including why Table 14 of 2407.08476 may not be reliable: questionable FLOPs and no architecture info).
- **Line 0, extra:** Add FlashAttention baseline (efficiency, performance) and comments on Mamba’s C++/CUDA optimizations.
- **Line 0, extra:** Results for long sequences (16/64 frames).
- **Line 0, extra:** Experiments on each token encoding global information and discuss Mamba’s bidirectional scan (each token encodes global information).
- **Line 0, extra:** Experiments: **add token importance block to TubeR**. And it shows that our token importance block can slightly improve TubeR's performance on our small scale experiments.

---

Thank you again for your careful consideration and support!

Best regards

---

### Note · Authors · 2025-08-12

Dear AC and Reviewers,

We are sincerely grateful for the constructive feedback and encouragement. Here, we present our *Author Final Remarks* in two sections: (1) the significance of our work and (2) key updates during rebuttal & discussion.

**(1) The significance of our work.**

This paper introduces **the first *pure* Mamba-based architecture for spatio-temporal video action detection**. MOGO achieves superior results in parameters, FLOPs, memory, and inference speed and competitive mAP. We believe this work opens a promising direction for efficient spatio-temporal action detection, therefore benefiting the broader AI community.

**(2) Key updates during rebuttal & discussion.**

**Mamba in video action detection** and the **token importance module** are the 2 core innovations of our paper. For clarity, we reorganize all *rebuttal & discussion* updates based on the innovations, followed by some minor paper quality improvements:

**How the rebuttal & discussion strengthen *innovation 1: Mamba in video action detection*:**
- Add AVA results, the first reliable Mamba evaluation across three benchmarks (JHMDB, UCF101-24, AVA).
- Test longer sequences (16/64 frames), showing scalability from Mamba’s linear design.
- Add FlashAttention to compare a *linear Transformer* with our linear Mamba structure.

**How the rebuttal & discussion strengthen *innovation 2: the token importance module*:**
- Add random and RLE token selection, validating the rationale behind our token importance module.
- Apply it to Transformer-based TubeR, showing this module's transferability.
- Clarify the design of the module more clearly. As comprising 3 parts: (1) encoder to learn token importance, (2) decoder to select important tokens, and (3) a redesigned loss function to supervise the process.

**How the rebuttal & discussion strengthen *the overall quality of the paper*:**
- Add diagnostics (training loss, gradient norms).
- Add related work.
- Clarify key concepts: *keyframe* action detection (why retaining all tokens may degrade the performance), *only glances once*, etc.
- Improve formatting and other minor revisions.

**We really hope our work can earn your appreciation and thank you very much for your kind attention**.

Best regards,

The Authors

---

### Decision · Program_Chairs · 2025-09-17

**Decision:**

Accept (poster)

**Comment:**

This paper proposes a Mamba-based framework for video action detection. It includes a pure Mamba-based encoder-decoder with better computational efficiency than Transformers and a video token construction mechanism that selects the most relevant spatio-temporal tokens. The proposed method demonstrates good performance in terms of efficiency and action detection ability on the JHMDB and UCF101-24 datasets though the accuracy (mAP) is considerably lower than for transformer based method (TubeR) .

The reviewers found the experiments in the submitted paper to be not sufficient.During the rebut period, the authors were able to report on additional experiments suggested by the reviewers; these included results on the AVA dataset and tests with different pretraining datasets and variations on token selection.

This paper received only three reviews. Based on the discussion, two reviewers raised their ratings to borderline accept and one to accept. There was also discussion of whether the paper required so many changes that it amounted to a new submission. The authors did a good job of summarizing the changes so the ACs feel that the revised content has been reviewed already. The ACs recommend that the paper be accepted.